# Global patterns in functional rarity of marine fish

Isaac Trindade-Santos [1✉], Faye Moyes [1] & Anne E. Magurran [1]

Rare species, which represent a large fraction of the taxa in ecological assemblages, account for much of the biological diversity on Earth. These species make substantial contributions to ecosystem functioning, and are targets of conservation policy. Here we adopt an integrated approach, combining information on the rarity of species trait combinations, and their spatial restrictedness, to quantify the biogeography of rare fish (a taxon with almost 13,000 species) in the world's oceans. We find concentrations of rarity, in excess of what is predicted by a null expectation, near the coasts and at higher latitudes. We also observe mismatches between these rarity hotspots and marine protected areas. This pattern is repeated for both major groupings of fish, the Actinopterygii (bony fish) and Elasmobranchii (sharks, skates and rays). These results uncover global patterns of rarity that were not apparent from earlier work, and highlight the importance of using metrics that incorporate information on functional traits in the conservation and management of global marine fishes.

[1] Centre for Biological Diversity, School of Biology, University of St Andrews, St Andrews, Scotland, UK. ✉email: its5@st-andrews.ac.uk

Pioneering ecologists, including Charles Darwin[1] and Henry Walter Bates[2], realised the importance of documenting geographic patterns in numbers of rare species when explaining variation in biological diversity over space and time. Rabinowitz's influential paper[3] on seven forms of rarity showed that it is possible to quantify these rare taxa objectively based on information about their geographic extent, local population size, and habitat specificity. Rare taxa play key roles in underpinning ecosystem function[4–6], and have priority in conservation planning[7,8]. Progress towards the goal of identifying which parts of the Earth support concentrations of rare species is greatest in terrestrial systems[9,10], with the geography of rarity in the oceans less well charted (but see Connolly, et al.[11] and Grenié, et al.[7] (for the tropics only)). To date, most evaluations of rarity have, sensu Rabinowitz, focussed on taxonomic diversity. Yet, as is now evident, biodiversity is a multi-faceted concept, and it is vital to incorporate information on ecological function in biodiversity assessments[12,13] and in conservation[14,15]. New rarity metrics[16,17] make possible an integrated assessment of rarity that takes account of the extent to which species exhibit rare combinations of traits, as well as being geographically restricted while a null model approach can be used to gauge whether such estimates of rarity levels are in excess of an expectation based on species richness. Here we identify concentrations, towards higher latitudes and in coastal systems, of bony and cartilaginous fish species that are rare in both taxonomic and functional dimensions, and document some mismatches between these rarity hotspots and the Marine Protected Areas of the world's oceans.

Our approach to measuring rarity draws on elements of taxonomic and functional biodiversity. First, we use the distribution data of all marine fish species (11,961 Actinopterygii and 866 Elasmobranchii) in AquaMaps[18] and allocate these to 2° grid cells in order to calculate a key facet of rarity, namely species geographic restrictedness[3] (see Supplementary Fig. 1 for workflow and Supplementary Fig. 2 for geographic restrictedness calculation). We then examine eleven ecological and biological relevant traits (7 continuous and 4 categorical traits, drawn from FishBase[19], representing key environmental, morphological, life history and reproductive variables) for each species, and use these data to quantify how functionally distinctive and unique each species is[16] (see Supplementary Fig. 2 for functional distinctiveness and uniqueness calculations). We define rare taxa as those that occur in the top (i.e. most rare) quartiles[20] of both restrictedness and distinctiveness distributions. Our measure of rarity thus reflects both taxonomic rarity (geographic restrictedness) and functional rarity (global functional distinctiveness and functional uniqueness), i.e. facets of rarity conceptualized respectively by Rabinowitz[3] and Grenié, et al.[16]. Based on previous findings that the tropics are the hotspots for marine species richness (taxonomic diversity) but have high levels of functional redundancy, e.g. species playing similar ecological roles[21,22] we expect to find low functional rarity in the tropics. Our analyses were undertaken independently for bony fish (Actinopterygii) and cartilaginous fish (Elasmobranchii). Moreover, our investigation was conducted separately for each Coastal System and High Seas in seven Oceanic Regions (14 different systems in total) (see Supplementary Fig. 3). That is, for each system, we assessed species rarity based on the distribution and functional traits of species present in that system. Finally, we repeated all the analyses using rarity defined by functional uniqueness[9] (instead of functional distinctiveness) to assess the generality of our findings.

## Results and discussion

As Fig. 1 shows, we detect substantial heterogeneity in the distribution of rarity in the world's Coastal Systems. Rarity is most striking in higher latitudes and coastal ecosystems. Hotspots of rarity occur on the East coast of Asia, Northeast and Southwest Australia and Southern Africa, with some concentration at the estuary of the Amazon River in Brazil and Central America for the bony fishes (Actinopterygii), (Fig. 1a), and on the East coast of Asia, the Southwest coast of Australia, South and West coasts of Africa, Central America and the UK for the cartilaginous fishes (sharks, skates and rays—Elasmobranchii) (Fig. 1c). When analysing the results for the High Seas we found more concentrations of rare species for both groups of species in the Atlantic and Eastern Pacific Oceans (Fig. 2a for bony fish and c for cartilaginous). These patterns are likely, to some extent, to reflect variation in species richness, which in line with other taxa[20,23] is highest in the lower latitudes (see Fig. 3a, b which depict species richness variation across latitude for bony and cartilaginous species respectively). However, when we control for species richness by comparing the observed rare species with the null expectation (see *Null Model* section at the Methods), our finding that high latitudes and coastal regions are associated with levels of rarity higher than expected by chance are reinforced (see Figs. 1 and 2b, d excesses (red) and deficits (blue) in rarity, relative to the null, and Fig. 3c–f for latitudinal variation in rarity, again as evaluated against the null). The hotspots identified in Figs. 1 and 2a, c remain, and are supplemented by a much larger sweep of coastline (Fig. 1), particularly in the Asia and Australasia, Africa, Red Sea (mainly bony fish), Amazon Estuary, Central America and UK (mainly cartilaginous fish) (Fig. 1b, d). Moreover, there is a marked concentration of rarity, in excess of expectations, at higher latitudes (Fig. 3, from c to f).

When we compared these results (Figs. 1 and 2, which employ the functional distinctiveness metric) with an alternative method of calculating functional rarity, namely functional uniqueness (Supplementary Fig. 4 (Coastal Systems) and Supplementary Fig. 5 (High Seas), see Supplementary Fig. 2 for indices differences) the conclusions were consistent. Interestingly, as the Supplementary Fig. 4 (Coastal Systems) and Supplementary Fig. 5 (High Seas) show, many of the same hotspots of rarity were identified, albeit with some differences along the South-western African Coast, Southern Indian Ocean, Red Sea (for bony fishes (Supplementary Fig. 4b)), and in Central America and South-eastern North America (Supplementary Fig 4d) for cartilaginous fish.

All analyses presented here used the probability of occurrences provided by the AquaMaps database >0.9 (see *Occupancy Data (Assemblage Matrix)* section in Methods for more details). As a further check, we repeated the investigations using probabilities >0.7 and >0.5. In all scenarios, our findings were consistent (see Supplementary Fig. 6a to d for probability >0.7 and Supplementary Fig. 6e to h for probability >0.5).

Distance decay plots, using the Northwest Pacific Ocean as an example (Supplementary Fig. 7) provide reassurance that spatial autocorrelation is not obscuring the results we report. In addition, we performed a sensitivity analysis to test if the environmental traits "mean temperature preference" and "maximum depth", along with the categorical traits (reproductive guild, body shape, swimming mode and position in water column) influenced our results. We found that even when we remove these traits from the analysis our results are consistent (Supplementary Fig. 8a to d). This supports our argument that the global patterns of rarity we uncover are not simply a response to traits influenced by local features. In conjunction with a trait correlation analysis (Supplementary Fig. 9a and b), this shows that no single trait was unduly influential on the patterns we found.

Do these patterns of rarity reflect heterogeneity in habitat use by the rare taxa? To explore this we compared the frequencies of "habitat specialization" categories (i.e. bathydemersal, bathypelagic, benthopelagic, demersal, pelagic neritic, pelagic oceanic and reef

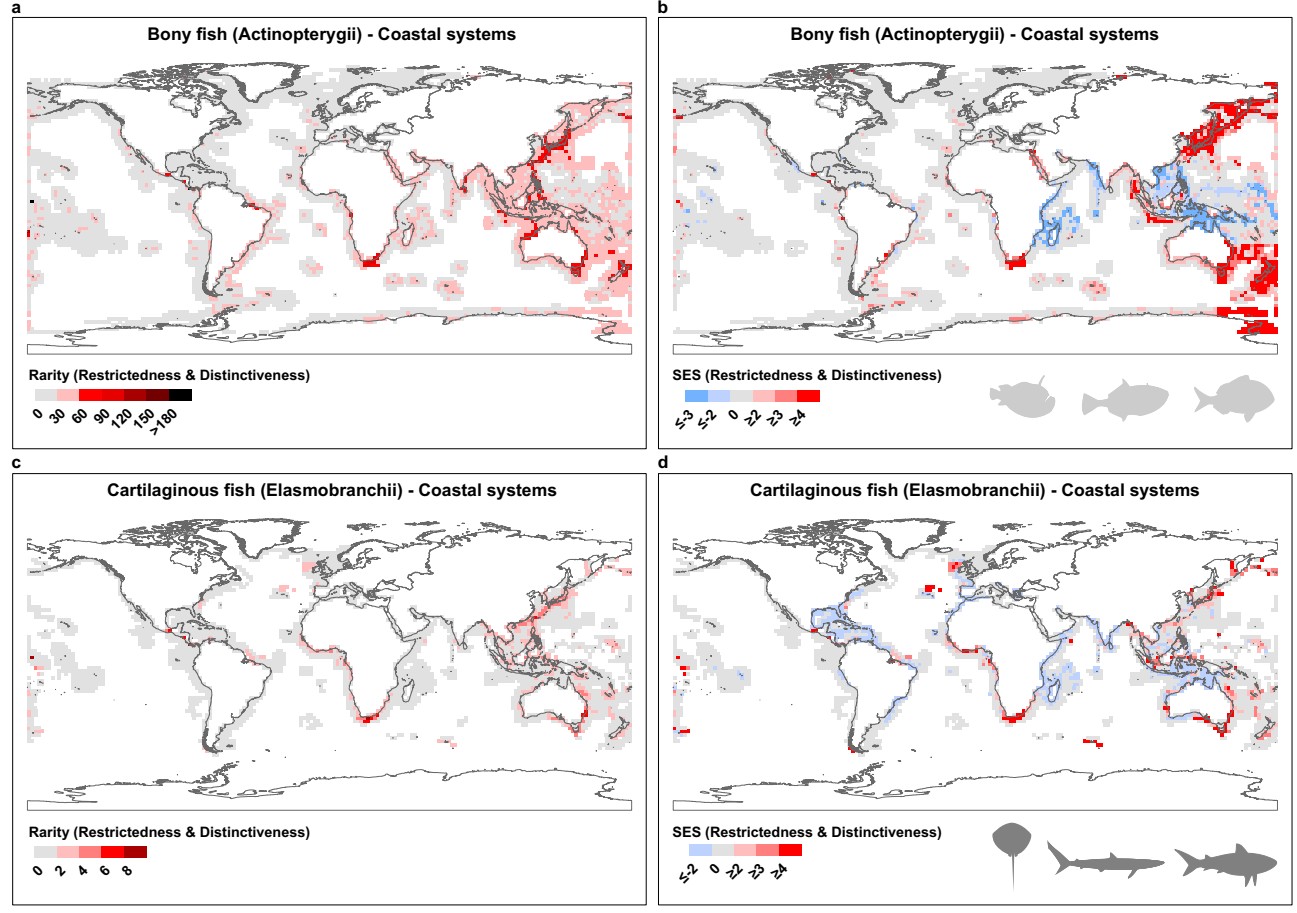

**Fig. 1 Global biogeography of rarity for bony fishes (a–Actinopterygii) and cartilaginous fishes (c–Elasmobranchii) across Coastal Systems.** The functional index used here was distinctiveness (see Supplementary Fig. 1 for workflow and Supplementary Fig. 2 for index details). Plots **a** and **c** illustrate the numbers of rare species found in each 2° grid cell (species that are both rare **taxonomically** and **functionally** (**distinct**)). Plots **b**, **d** shows the Standardized Effect Sizes (SES), where red shaded cells represent an excess of rare species higher than expected by chance and blue cells represent fewer rare species than expected. For results using functional uniqueness, see Supplementary Fig. 4.

associated) in the rare and non-rare species and found evidence of differences in their association with different habitat types (Supplementary Fig. 10). For example, in the Northwest Pacific Coast most non-rare species are classified as reef associated (Supplementary Fig. 10a (pale green)), while rare species are mostly classified as demersal (Supplementary Fig. 10a (dark green)).

It is possible that a sampling bias towards records in coastal waters could partly explain our findings. However, we note that regions of the earth that are typically under-recorded in biodiversity surveys[24], such as South America and Africa (but see Floeter et al.[25,26], and Pinheiro et al.[27,28]), are identified in our analysis as important repositories of rarity, while marine areas that have been intensively monitored, such as the coasts of NW Europe[29–31] are not. Additional analyses showed that sampling effect has no influence on the identification of rarity hotspots (see Supplementary Analysis and Supplementary Fig. 11).

The striking concentrations of rarity uncovered by our analysis are incompletely represented by existing Marine Protected Areas (MPAs). Here we defined cells exhibiting rarity as those where SES > 2. We found mismatches between the location of 2° grid cells with these high levels of rarity and MPAs (see Fig. 4a and b, bony and cartilaginous fish respectively). This was true for both bony fish (matches in 47% of MPAs (Fig. 4a)) and cartilaginous fish (matches in 63% (Fig. 4b)). Thus, on average, only around half of the localities in the oceans with notable levels of rare fish are protected by MPAs.

AquaMaps[18,32] provided us with a unique opportunity to use occurrence data for measuring rarity based on occupancy data. We treated species occurrences in each of the two-degree grid cells as a proxy for rarity and used this information to distinguish between rare species (highly restricted species) and common species (widely occurring species). The global coverage of AquaMaps is an advantage here. There are, however, some caveats associated with our approach. For example, the data provided by AquaMaps include not only actual observations but also probabilities of occupancy based on combinations of sightings and internal algorithms. Moreover, AquaMaps is based on summed information rather than being a representation of the species present at a given point in time. Additionally, the nature of our data means we cannot calculate shifts in rarity on a temporal scale or with any weighting by abundances. We also appreciate that when abundance and/or biomass data are available, another taxonomic rarity metric, namely Taxonomic Scarcity[16], could be used. Our decision to focus on restrictedness was shaped by the fact that we are using geographical information from all described fish available in AquaMaps, and that this database provides occupancy patterns based on different probabilities of occurrences.

Our analysis highlights the new insights gained from an integrative approach to quantifying rarity. It shows that using other proxies of biodiversity importance, such as hotspots of richness, and MPAs do not effectively capture the regions of the ocean that

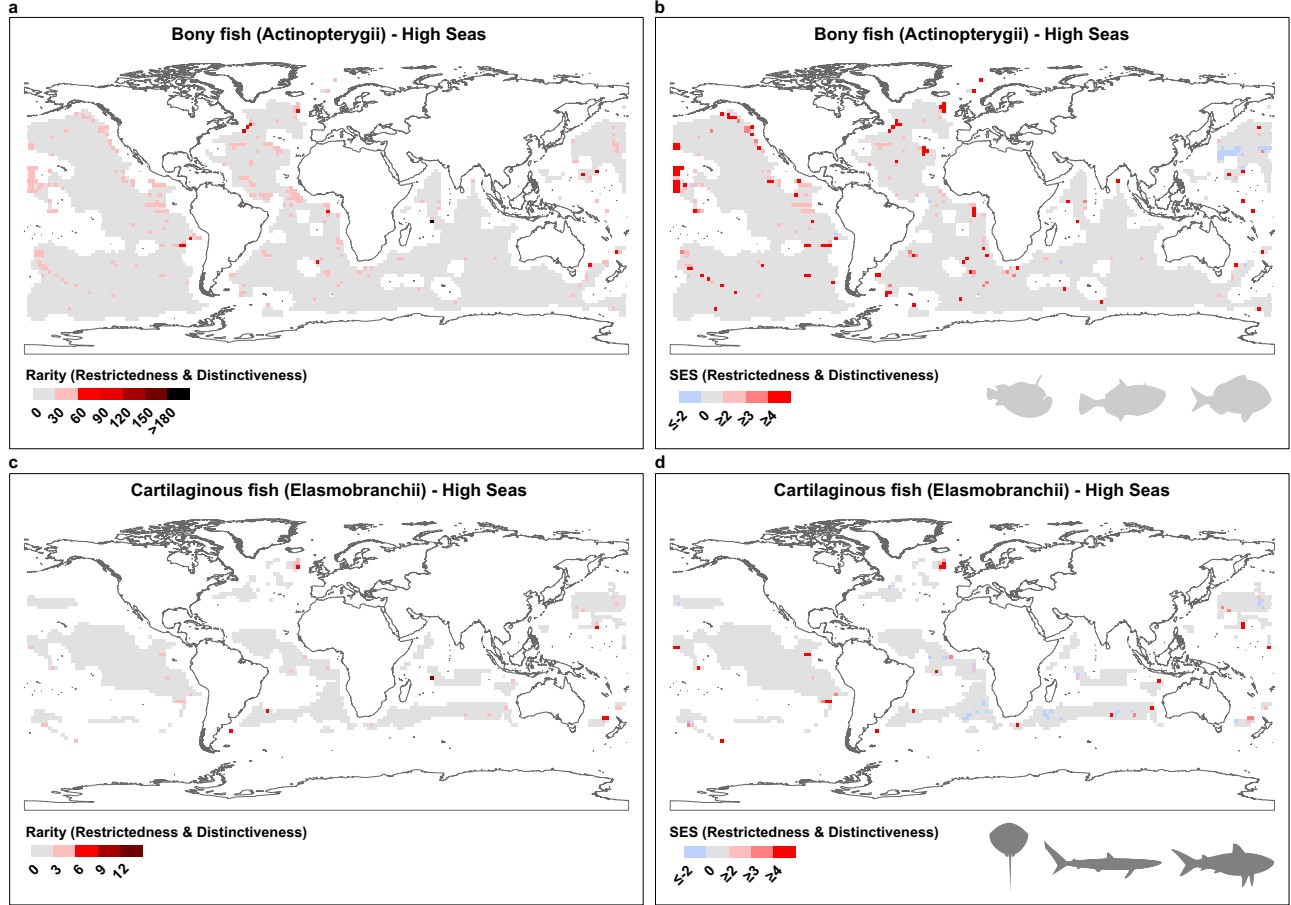

**Fig. 2 Global biogeography of rarity for bony fishes (a–Actinopterygii) and cartilaginous fishes (c–Elasmobranchii) across High Seas.** The functional index used here was distinctiveness. Plots **a** and **c** illustrate the numbers of rare species found in each 2° grid cell (species that are rare both **taxonomically** and **functionally** (**distinct**)). Plots **b** and **d** show the Standardized Effect Sizes (SES). For results using functional uniqueness, see Supplementary Fig. 5.

support concentrations of rare taxa. Indeed, it is higher latitudes, and in particular coastal areas, where rare taxa are most evident. May[33] observed that the oceans are richer in phyla, but poorer in species, than terrestrial ecosystems. He attributed this to both the long evolutionary history of marine life, and the relative lack of spatial heterogeneity in the oceans. Both factors are likely to contribute to the patterns we uncovered. On the one hand the ancient phylogenetic roots of marine fish provide opportunities for trait innovations to arise[34]. On the other, the spatial structure of coastlines and inshore waters is likely to foster local adaptation[35,36]. Similarly, Rabosky[34] found faster speciation rates outside the tropics, particularly at higher latitudes; it is possible that those regions acts as centres for evolutionary and functional novelty (as shown in our study).

The results we report demonstrate the importance of an integrative approach in conservation to protect rare species that exhibit distinct functional traits[37] and make important contributions to ecosystem functioning[4,10,38]. The advantages of such an approach are already clear. For example, Hidasi-Neto et al.[14] used the combination of different facets of biodiversity (species distribution, functional and phylogenetic distinctiveness) and the IUCN species threat status of terrestrial mammals to highlight the importance of the effectiveness of conservation efforts by focusing on regions with high ecologically and evolutionarily distinct mammal species. However, rare taxa are not necessarily conservation priorities at present. For instance, rare terrestrial mammals and bird species are not covered by protected areas at a global scale[10] even though elevated extinction risk is linked to high ecological distinctiveness[39].

Concentrations of rare coral reef species also have mismatches with MPAs[7].

Human exploitation, combined with environmental change, is threatening rare taxa with important functional roles, including marine megafauna[8]. In particular, we note that the ongoing global decline of sharks, which display the highest functional distinctiveness amongst the fishes[8], will contribute to a substantial loss of functional diversity in marine megafauna[8,40,41]. Fishing pressure and climate change are known threats to fish functional diversity[42] and rarity, as a recent study of the North Sea shows[43].

As the vision for the United Nations Decade of Ocean Science for Sustainable Development makes clear, there are still substantial knowledge gaps concerning the status and trends of marine biodiversity. Our paper helps fill one of these knowledge gaps and emphasises the importance of adopting an integrated approach to evaluating, and protecting, the biodiversity on which we depend.

## Methods
We used AquaMaps[18,32] to obtain species occurrence data, and extracted eleven ecologically and biologically relevant traits (seven continuous and four categorical) from FishBase[19]. All calculations were done separately for the two group of fishes (bony fishes and cartilaginous fishes). Our workflow is in Supplementary Fig. 1.

### Step 1- Input
*Occupancy Data (assemblage matrix).* We extracted occupancy data from AquaMaps[18,32]. This online database provides half-degree grid cell species occurrences based on data from GBIF[44] and OBIS[45] complemented with information from FishBase[19] and SeaLifeBase[46]. AquaMaps gives probabilities of the occurrence

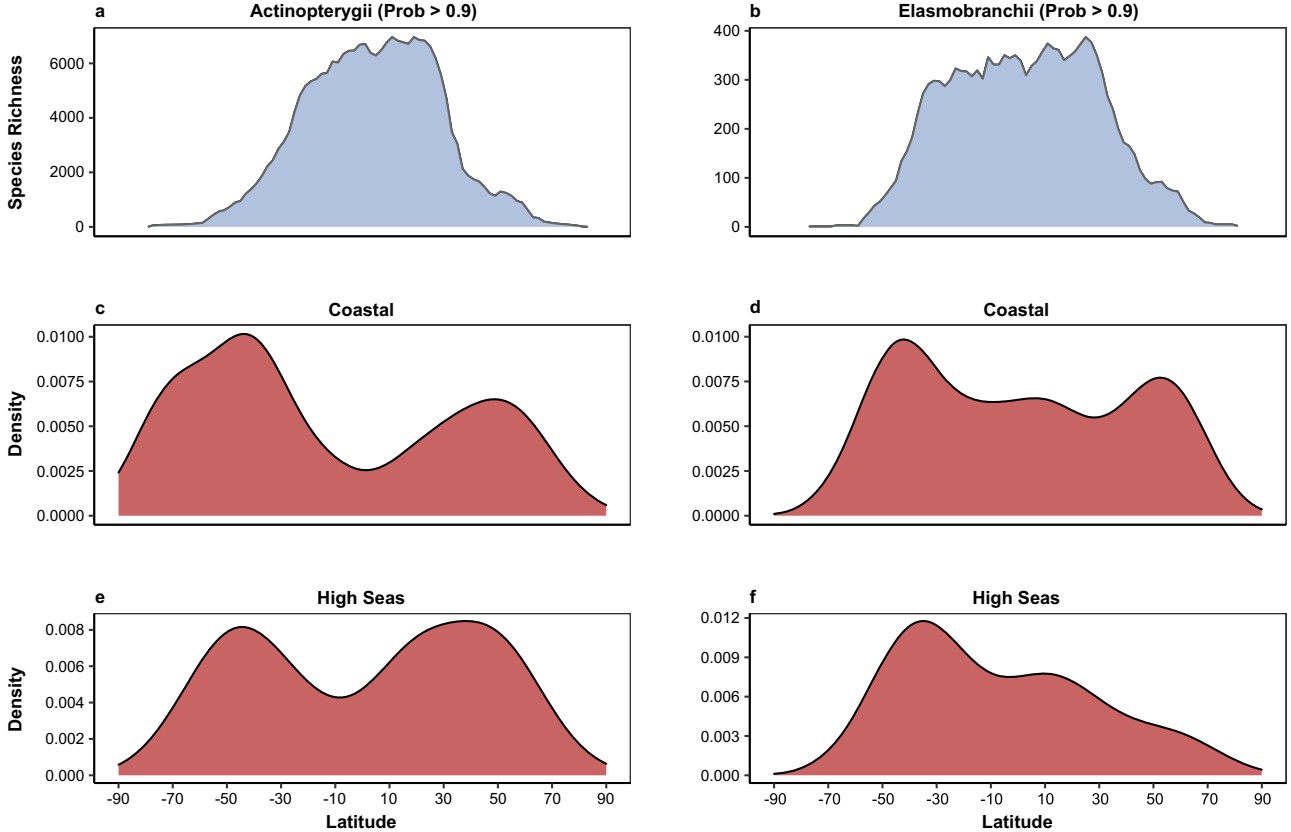

**Fig. 3 Illustration of the latitudinal biogeographic distribution of different metrics: species richness (a and b), positive SES ( > = 2) based on distinctiveness (from c to f). a, b** plot the distribution of species richness in relation to latitude for Actinopterygii (left plots) and Elasmobranchi (right plots) respectively. Plots from **c** and **d** represent **Coastal Systems**, while **e** and **f** show the SES distribution for the **High Seas**. Here we use the **probability of species occurrences higher than 0.9** (see Supplementary Fig. 6 for the latitudinal results found using probability > 0.7 and > 0.5).

of a given species between 0 (based on the environmental envelope indicating there is no chance of finding the species in that grid cell) to 1 (highest probability to find the species in that grid cell)[32].

These occurrence data are then combined with an algorithm implemented by AquaMaps using "estimates of environmental preferences with respect to depth, water temperature, salinity, primary productivity, and association with sea ice or coastal areas". For more information see Kesner-Reyes, et al.[32].

In our analysis, we selected occurrence data (the presence of a given species in a certain half degree grid cell) with a probability >0.9. To ensure our results were not simply an artefact of using a high probability of occurrence we also examined probabilities higher than 0.7 and 0.5. Further analyses were repeated independently for each of these probabilities. After this initial step, we allocated each half-degree grid cell to a 2° grid cell. We created the 2° grid cells using features available in ArcGIS[47]. These occupancy data also allowed us to compute the species richness of each 2° grid cell.

We applied all analyses described here (see Supplementary Fig. 1) separately for the Coastal Systems and High Seas of each of the Seven Oceanic Regions (see Supplementary Fig. 3). This gives us seven Coastal Systems and seven High Seas, a total of 14 assemblage matrices for each of the probabilities (probability > 0.9, 0.7 and 0.5).

*Trait Data Compilation (trait matrix).* Our final occurrence data (each of the 14 systems) were split between marine bony fishes (11,961 Actinopterygii species) and cartilaginous fishes (866 Elasmobranchii species). For each species in both groups of fish we assembled biologically and ecologically relevant traits from the most recent version of the FishBase database[19]. This gave us seven continuous and four categorical traits that are largely uncorrelated with one another, (see Supplementary Fig. 9a and b):

Environmental Traits: (1) *Position in Water Column*—The vertical position of a species indicates its feeding habitat[19] and its influence on the process of transferring nutrients through the water column[48,49]. This trait has 8 categories. (2) *Maximum Depth (m)*—This trait reflects the environmental conditions that each species occur[27]. (3) *Mean Temperature Preference* (°C)—Thermal variations in preferences indicates the species tolerances to changes in temperature[50–52].

Life History Traits: (4) *Growth (k)*—This coefficient parameter (k = 1 year$^{-1}$) is derived from the von Bertalanffy growth function ($L_t = L \infty$ (1-exp($^{-K(t-t0)}$))).

Faster growth rates are associated with higher k values[19]. (5) (*Q/B*)—this trait represents the proportional ratio between food consumption (Q) and biomass (B) and can be used as a proxy for trophic interactions, evidencing the flow of energy in the system[53,54]. (6) *Trophic Level*—the position of a given species in the food chain is expressed as its trophic level and as discussed by Froese and Pauly[19] can be assessed as the amount of "energy-transfer steps to that level". The trophic level also gives information on interactions between species, for example predator-prey and trophic cascades[55].

Morphological Traits: (7) *Body Shape*—The body shape of a species relates to ecological behaviours, such as migration patterns[56]. We divided this trait into 38 categories (see Supplementary Table 1). (8) *Swimming Mode*—the mobile strategy adopted by fish species has a direct relationship with ecology and behaviour[57]. Following FishBase we used 12 different swimming modes (see Supplementary Table 1) based on anatomical and morphological features; these traits provide information on the functional role that each species plays.

Reproductive Traits: (9) *Generation Time*—defined by FishBase as "the time period from birth to average age of reproduction". (10) *Length of First Maturity (mm)*—body length when around 50% of a given species becomes mature[58,59]. (11) *Reproductive Guild*—15 categories of reproductive guild as defined by FishBase (see Supplementary Table 1). Reproductive traits have a direct influence on population dynamics and species resilience[60,61], and are therefore commonly used in fisheries management[62].

These traits were selected for their ecological and biological relevance as described above. We tested the correlations of the traits to ensure complementarity, and as shown in the (Supplementary Fig. 9a and b), these traits are largely uncorrelated. We also took into consideration the data gaps inevitable in a large data set such as FishBase (traits were selected if a maximum of 30% of data were missing). To overcome this limitation we applied random forest algorithms to fill the missing traits[63], by using the package "missForest"[64].

**Step 2—Rarity indices**

*Species restrictedness.* We calculated species restrictedness ($Res_i$) by dividing the species geographical extent (GE = number of 2° grid cells that a species occurs in based on the assemblage matrix compiled from AquaMaps) by the total number of grid cells (TOT), minus one (see Supplementary Fig. 2). We scaled the values

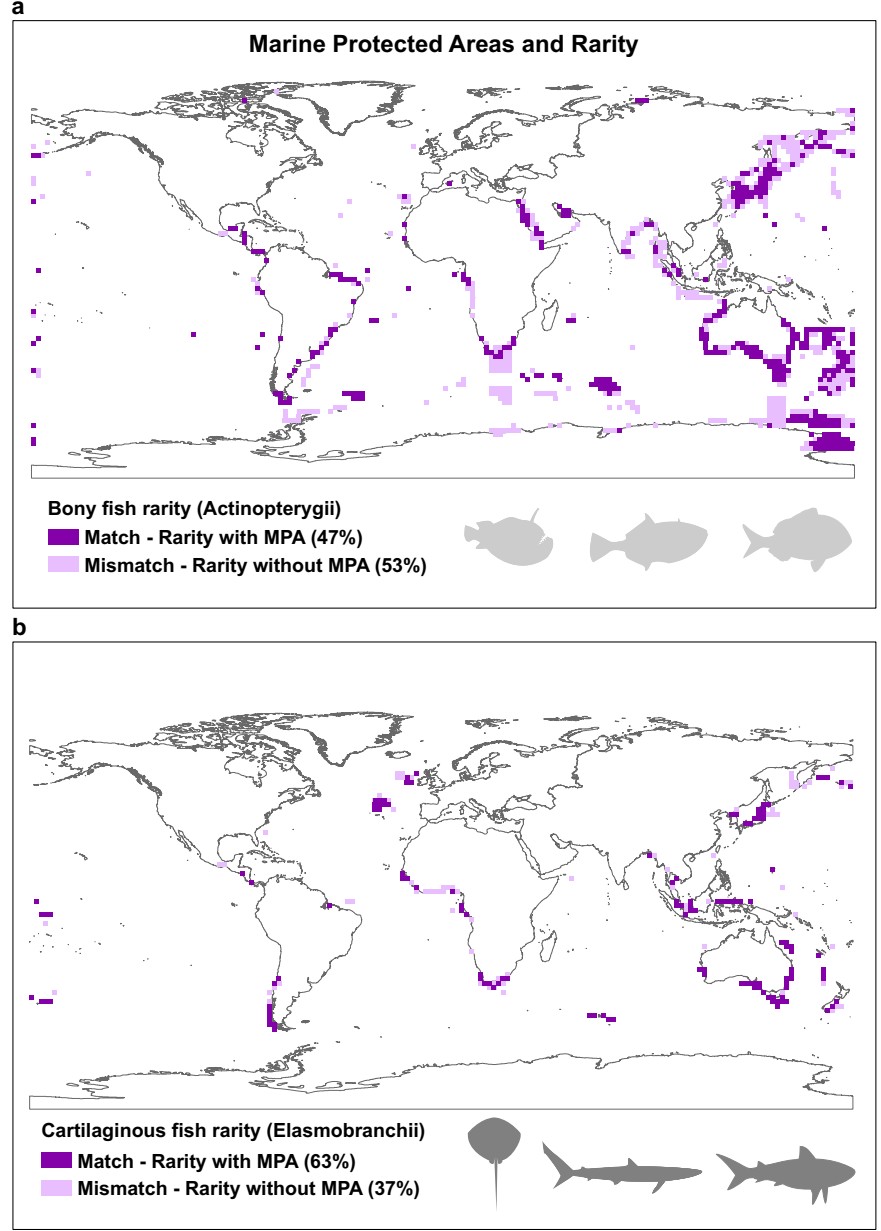

**Fig. 4 Congruence and mismatches between MPAs and rarity hotspots.** We illustrate the results for the Actinopterygii **a** and Elasmobranchii **b** for the Seven Oceanic Regions. In this example, the grid cells presenting Standardized Effect Sizes higher than 2 with the presence of a Marine Protected Area are represented as dark purple, meanwhile the soft purple are the grid cells with SES > 2 and no MPA. The figure shows that 47% of grid cells identified with high concentrations of rarity coincide with MPAs for bony fish **a** and 63% for cartilaginous fish **b**.

between 0 (species occurring in all 2° grid cells) and 1 (the most restricted species). We used the function "*restrictedness*" in the "funrar" package to do this calculation[16].

*Functional distinctiveness.* Functional distinctiveness ($Dis_i$) quantifies the level of dissimilarity in trait combination between species[16,22] (see Supplementary Fig. 2). This index is the average of functional distance of a given species compared with all other species in the assemblage[16].

We calculated how distinct or common each species is by using the function "*distinctiveness_global*" available in the "funrar" package[16]. We then scaled the values found between 0 (species with common combinations of traits) and 1 (species with the most dissimilar combination of traits). This analysis was conducted using presence/absence data.

*Functional uniqueness.* Functional uniqueness quantifies the level of species isolation in the multidimensional functional space[16,17]. This index is calculated by quantifying the distance of each species in relation to its nearest neighbour[16]. This mathematical approach applied to multidimensional functional space was adapted

from the mean nearest neighbour distance developed initially to calculate the phylogenetic distance between species[65] (see equation descriptions at the Supplementary Fig. 2).

**Step 3—Selecting rare species & Step 4—Rarity biogeography**
*Quartile analysis.* We examined the distribution of values for species restrictedness and functional distinctiveness (or species restrictedness and functional uniqueness) and used the quartile criterion (performed using the base R function "*quantile*" from the package "stats" in R Core Team[66]) as proposed by Gaston[20] to identify the rare species. By this definition the species considered rare lie in the top quartile of both metrics (i.e. values between 0 (less restricted) and 1 (more restricted)). We next assigned the observed number of rare species (Step 4), as defined above, to each 2° grid cell. The analysis was undertaken separately for Actinopterygii and Elasmobranchii.

**Step 5—Null model**. Does the number of rare species in a given grid cell differ from the null expectation? To answer this question we applied a null model

approach based on the curveball algorithm[67]. This algorithm keeps constant the total number of species (rare + non-rare) and the number of grid cells that each species occurs. It then randomizes the presence and absence of all species following these thresholds. We ran the model for 2000 iterations; in each loop it randomizes the occurrences of all species, identifies where the rare species are falling and then counts the total number of rare species in each grid cell.

To quantify how the observed number of rare species differ from the null expectation we then use Standardized Effect Sizes (SES) as follows:

$$SES = (X - Y)/Z$$

X as the number of rare species observed in each grid cell,
Y as the average of rare species found from the null model after 2000 interactions and
Z as the standard deviation from Y.
A positive SES indicates more rare species than would be expected by chance and a negative SES fewer than expected.

We are using 14 different systems (7 Coastal Systems and 7 High Seas (see Supplementary Fig. 3)) for 2 groups of organisms (bony and cartilaginous fish), 2 functional rarity indices (distinctiveness and uniqueness), and using 3 different probabilities of occurrences (prob. >0.9, >0.7 and >0.5).

We then have the following "roadmap":

i. Scales—7 Coastal Systems and 7 High Seas.
ii. Groups—bony and cartilaginous fish.
iii. Indices—distinctiveness and uniqueness.
iv. Probability of occurrences—>0.9, >0.7 and >0.5.
v. Total—168 independent cases analysis (each having its own assemblage and trait matrices.

Therefore, as mentioned above, the null model ran for 2000 iterations in each of those independent cases. The final matrices from these initial steps contain grid cells as rows and as columns we have the raw number of rare species along with the SES values for each. These matrices were important to map our results.

**Step 6—Mapping the results**. After the above steps (and using the matrices with the results), to visualise the results for Coastal Systems we plotted the geographic distribution of rarity, measured using the observed number of rare species, based on species Restrictedness and functional Distinctiveness using Fig. 1a, c, and the results from the SES using Fig. 1b, d. Meanwhile in Fig. 2 we constructed the same plots for the High Seas. The complementary results of the alternative approach using species Restrictedness and functional Uniqueness are shown in Supplementary Fig. 4 (for Coastal Systems) and Supplementary Fig. 5 (for High Seas).

The flow chart in Supplementary Fig. 1 provides step by step details of what was done for each of the 168 independent cases explained at the "roadmap" above. The comprehensive list of all rare fish species found for each system is available in Supplementary Table 2.

**Further analyses**
*Latitudinal rarity biogeography*. We then produced the density plots of the positive SES values (using the function *geom_density* from the package ggplot2[68]) to further understand these patterns in relation to latitude (Fig. 3, from c to j). These were compared with the latitudinal gradient of species richness (Fig. 3a, b). The main text discusses results focused on rarity measured using the probability of occurrence higher than 0.9 (Fig. 3). We also examined density of positive SES values across the latitudinal distribution using the probability of occurrences higher than 0.7 and 0.5 (see Supplementary Fig. 6a–d for probability >0.7 and Supplementary Fig. 6e–f for probability >0.5).

*Spatial autocorrelation*. We constructed distance decay plots to examine spatial autocorrelation, and fitted a quantile regression to these relationships. The results are illustrated in Supplementary Fig. 7, which shows the distance decay calculated by pairwise differences (Supplementary Fig. 7a—Coastal Systems and b—High Seas for bony fish, and Supplementary Fig. 7c—Coastal Systems and d—High Seas for cartilaginous fish) between a given grid cell and all other grid cells present in the Northwest Pacific Ocean. These plots provide reassurance that spatial autocorrelation is not obscuring the results we report.

*Sensitivity analysis*. We performed a sensitivity analysis to ensure that the environmental traits "Depth" and "Mean Temperature Preference" had no major influence on determining the level of distinctiveness and uniqueness of the species. We did this by excluding each trait in turn from the analysis (each of those were removed individually and a third time without both together) and compared the results with the full analysis. We found strong correlations in rarity estimates in all cases (see Supplementary Fig. 8a, b (for bony fish (distinctiveness and uniqueness) respectively, c and d (for cartilaginous fish (distinctiveness and uniqueness)).

*Trait correlation analysis*. We tested the correlation between traits to ensure that those were largely uncorrelated, as shown in Supplementary Fig. 9a, b.

*Supplementary analysis*. We tested the possible influence of sampling effect on the rarity hotspots observed by creating random fill matrices and comparing those with the observed matrices from four scenarios: Northwest Pacific Coast (bony and cartilaginous fish species) and Southwest Pacific Coast (bony and cartilaginous species). The subsequent results showed no evidence of sampling effect (see Supplementary Fig. 11).

*Mapping marine protected areas (MPAs)*. We used the MPAs shapefiles provided by the UNEP-WCMC and IUCN[69] to measure the level of congruence between marine protected areas and hotspots of rarity. The distances between each MPA and the centroid of each grid cell were calculated using the spatial analysis tool in ArcGIS (the unit of the distance calculated is in decimal degrees). We then assigned each MPA to its nearest 2° degree grid cell centroid (the distance cut point used was < 0.75 decimal degrees (the distance from a given MPA to a grid cell centroid)). We plotted these global spatial patterns from the 2° grid cells indicating either congruence or mismatches between Marine Protected Areas (MPAs) and Rarity Hotspots (species rare in both dimensions of biodiversity; taxonomically—highest restrictedness and functionally—highest distinctiveness) (Fig. 4a and b, bony and cartilaginous fish respectively).

*Habitat specialization*. All species were classified according to their position in the water column (bathydemersal, bathypelagic, benthopelagic, demersal, pelagic neritic, pelagic oceanic and reef associated (as categorized in FishBase[19])); here we are using this trait as a proxy for "habitat specialization". We then used a G test, and Cramer's V (using the functions *GTest* and *CramerV* from the package *DescTools*[70]) to compare the frequency distribution in habitat specialization between rare and non-rare species (see Supplementary Fig. 10 for all frequency distributions and statistical results).

**Forms of functional rarity classification scheme**. In their 2017 paper Violle et al.[17], suggested 12 forms of functional rarity. We believe that the approach applied here is similar to the classification scheme they described as: "Rare traits irrespective of the scale and the species pool". The authors pointed to two possible extremes: rare traits (exhibited by range-restricted species) and common traits (supported by many widespread species). In this case, our approach identifies species that are both geographically restricted within each of the 14 systems (coastal and high seas systems) and present a distinct (or unique) combination of traits. Our approach to the classification of rarity differs slightly, however, in that we follow Gaston's approach[20] of quantile distribution as illustrated in Supplementary Fig. 1, step 3 QUARTILES.

**Reporting summary**. Further information on research design is available in the Nature Research Reporting Summary linked to this article.

## Data availability
The datasets generated during and/or analysed in this study are available in the Research Portal of the University of St Andrews repository, https://doi.org/10.17630/397bc872-f7de-4ded-9ed8-4f734c11b14a[71].

## Code availability
Code is provided in the Supplementary Methods section of the Supplementary Information file.

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

## Acknowledgements

I.T.S. thanks CAPES (*Coordenação de Aperfeiçoamento de Pessoal de Nível Superior* -Coordination for the Improvement of Higher Education Personnel), process number: #88881.129579/2016–01 (Finance Code 001) for a PhD scholarship. A.E.M. thanks the Leverhulme Trust (RPG-2019–402) for support. We thank the FishBase team (The Philippines and Nicolas Bailly) and the AquaMaps team (Cristina Garilao) for making their data available, the "Fish Lunch" group at the Centre for Biological Diversity (University of St Andrews), Daniel Pauly for their helpful comments, Ylenia Cascone for implementing the null model in Python and Piter Nascimento West for fish drawings. I.T.S. thanks Kátia de Meirelles Felizola Freire and Sidney Feitosa Gouveia for earlier guidance, research opportunities and inspiration.

## Author contributions

I.T.S. developed the concept of the paper in conjunction with A.E.M and F.M. All authors contributed to the analyses and to the writing of the paper.

## Competing interests

The authors declare no competing interests.
