## [Peer Review File · Nature Communications]

Reviewer comments, first round review:

Reviewer #1 (Remarks to the Author):

Thank you very much for giving me the opportunity to review the paper "Biogeography of rare fish in the world's oceans". The Authors highlight how current estimates of rarity focus on species' geographic ranges. As an alternative, they propose to measure rarity by looking not only at species range but also at species functional distinctiveness and uniqueness (where rare species have a trait combination that does not occur in other species). After obtaining such a measure for a large set of fish species (by using information on various life history and ecological traits obtained from FishBase), they reprojected rarity at the global scale (at a relatively coarse resolution of 5 x 5 degrees), as the number of rare species identified according to their new criterion. To avoid obvious potential biases due to the expected relationship between overall local species richness and rare species' richness, they quantified local rarity by looking at the residuals of the overall rarity/richness relationship. They then explored geographical patterns of rarity, the overlap/mismatch between rarity hotspots and protected areas, and the effect of environmental variables on rarity. They also performed a sensitivity analysis to explore the impact of functional traits' choice on the rarity patterns.

They found that rarity is higher in coastal areas and higher latitudes, with various (quite large) areas of hotspots. Although not coincident, the pattern is relatively consistent for bony fish and cartilaginous fish. The Authors also found a widespread mismatch between rarity hotspots and protected areas.

I think the study is interesting and has value, but I have a few general conceptual and methodological concerns which make me cautious about the results and their interpretation.

1) Goal/novelty of the study

Reading the abstract and the Introduction, a reader has the impression that the Authors have developed a new concept of rarity going beyond species range (i) and then they have applied that for the first time to fish (ii) at the global scale (iii). Of these three (implicit?) claims, only the 3rd one is true. The idea of considering functional traits to estimate rarity is actually not new and very well discussed, for example, in this perspective paper:

<https://doi.org/10.1016/j.tree.2017.02.002>

The approach has been already applied to fish, with a very similar conceptual and analytical workflow:

<https://doi.org/10.1016/j.biocon.2018.08.011>

That paper examines functional (and evolutionary) rarity patterns for more than 2000 reef fish globally.

So the main novelty here is the application of the approach at a global scale not limited to the tropics. This should be clarified in the Introduction. A brief discussion of the previous studies/results would also be helpful to set up the scene. The title is quite generic and not informative about the paper's goals/findings.

2) Scale of functional rarity and choice of the functional/ecological traits

I am not 100% I understood this correctly, but my impression is that the Authors assessed rarity by looking at the complete set of fish species (separately for bony and cartilaginous fish). I suspect that this, combined with the choice of the environmental traits (especially maximum depth, mean temperature preference) and some adaptive traits (such as swimming mode, body shape), might create some circularity. That is, the measure of global rarity for a given species might partly reflect the availability of habitat/conditions suitable to the species. Then, I might expect to have more niches for rare species where the local environmental setting is rarer.

To make an extreme example, I can imagine that a specific set of environmental/ecological conditions can be found only in a few localities worldwide. Some species might be well adapted to

those particular conditions. Therefore, their range would be likely restricted to that locality, and their functional traits would also reflect the peculiarity of the environment. The Authors' approach would therefore identify those species as rare and the area where the species occur as a hotspot of rarity. This is fine, but I think that it would be quite important to discriminate between this kind of situations (where local rarity is a consequence of local environmental/ecological features) and a situation where rarity depends only on species. From a conservation perspective, this is very important (as the first case would call for management strategies focusing on the site; while the latter would require strategies focusing on the species).

The confounding effect of the role of the environment on species traits (and hence the potential link between global patterns of environmental variability and global patterns of rarity) could be possibly reflected in the large scale overall results. Especially, the main findings of more rarity in the coastal areas and at higher latitudes seem to me as potential consequences of this mechanism. Coastal environments clearly provide more environmental heterogeneity (and hence possibly more specialized habitats promoting the presence of rare species according to the Authors' metric) compared to the pelagic realm. The pattern of increasing rarity with latitude might be even more an effect of the analytical approach: by pooling all species together when calculating rarity, given the higher fish diversity of the tropics and the selectivity of the high-latitude environments, it is not surprising to find more rare species in temperate and cold-waters realms than in the tropics (which in turn drives the observed patterns). The Authors tried to control for this effect, but see my comments below on their approach.

One potential solution to the issue is that of examining patterns of rarity by splitting the global species pool into subsets based on some biogeographical/environmental criteria. The issue was considered also in the aforementioned paper about reef fish functional rarity ("We tested the influence of the species pool definition from which functional distinctiveness was extracted. To this aim, we split the cells into two realms—Atlantic and Pacific—and estimated species distinctiveness values separately in each realm").

Selecting a different set of functional traits not reflecting adaptation to specific habitat conditions could be another potential strategy, but I am not sure which traits I would use. Perhaps removing at least the mean temperature preference and/or the max depth could be one starting point, but I guess the issue is a bit philosophical as every single trait could be seen as an adaptation to some environment and I cannot see any strong rule or argument to identify traits that are more or less important to define a species rarity.

To this end, it would also be important to consider that the concept of rarity is a very relative one (while the Authors seem to present it as a quite absolute measure). In this regard, I think Table 1 in the perspective paper mentioned above (<https://doi.org/10.1016/j.tree.2017.02.002>) provides a valuable roadmap to the different possible types (12) of functional rarity. I am not suggesting that the Authors need to explore all the possible types of rarity, but at least they should clarify where their measure falls into the classification scheme.

3) Disentangling the effect of richness from rarity.

Considering the potential issues of circularity highlighted above, I appreciate that the Authors attempted to correct for the potential effect of species richness. However, I have a few concerns about their approach (looking at the residuals of the linear regression of richness vs. rarity). I could not find in the text or SI information on the actual regression (apologies if I missed it). Is the relationship really linear and well fitted by the regression? Otherwise looking at the residuals would not be very reliable. Grenie' et al. (in the global reef fish rarity paper referred to above, <https://doi.org/10.1016/j.biocon.2018.08.011>) used an explicit null model approach that seems much more flexible and robust to me (they compared the observed patterns with a set of randomized species/locality matrices obtained by reshuffling species occurrences while keeping fixed the number of occurrences per locality, and the number of localities where each species occurs; they assessed local rarity in terms of standard effect size from the comparison of observed vs. null rarity).

In addition, linking this aspect back with the circularity issue above, I think that the approach of looking at the residuals of the richness-rarity relationships might even worsen the issue. In fact, my intuition is that measuring richness on the global species pool would tend to identify rare

species in the areas with lower diversity (because of the more selective environment). However, due to the large global variation in species richness (with orders of magnitude differences in the number of species per 5x5 grid cell globally), this effect would tend to disappear/be reversed in the overall relationship between richness and rarity (which will be, I think/infer from the text, a positive relationship). Consequently, looking at the residuals will reinforce the expectation of rarity in areas with low diversity (looking at the issue from another perspective, I think using the OLS residuals is conceptually close to showing relative rarity, i.e. the fraction of rare species per total local richness; the impact of a rare species will be much higher in a locality with low diversity compared to a hotspot of diversity, which might explain the results in fig. 1 (with the usual hotspots of diversity showing very low rarity)).

(4) Coastal vs. pelagic

The low (almost none?) rarity in the pelagic realm might be the result of two aspects. The first one is the fact that most pelagic species have wide ranges (and the same issue applies in part to the results for sharks) and, by necessity, quite generalistic traits (not surprising in a heterogeneous environment, see point 2). Second, there is an obvious sampling bias when comparing pelagic vs. coastal habitats. The very limited knowledge about deep fish is clearly preventing us from making a fair comparison of rarity, as we cannot tell how many rare species are there at the ocean bottoms. I am not saying that the Authors should fill in this knowledge gap, I am just saying that treating coastal habitats in the same way as pelagic habitats is a bit unbalanced, and I would have probably not done that.

(5) Ranges vs. abundance

As previously shown (<https://www.nature.com/articles/nature12529>), accounting for species abundances is fundamental to explore patterns of diversity and rarity. Again, I am not saying that the Authors should do this here, but I think that it would quite important to at least acknowledge this (see also the importance of population abundance in the definitions of rarity in the above-mentioned Table 1 from Violle et al.). In particular, it is true that many pelagic species are still widespread, but the populations of many of those are dramatically declining. Same for sharks.

Reviewer #3 (Remarks to the Author):

MS "Biogeography of rare fish in the world's oceans". The authors did a great job in gathering big datasets to address spatial patterns of rarity across the oceans and checking the superimposition between rarity and the location of marine protected areas (MPAs). While I believe this manuscript is of interest to the audience of Nature Communications, I identified a set of issues that prevent me from accepting this manuscript in its present form.

Rabinowitz's concept of rarity embraces three facets: range size, abundance, habitat specialization. In this study, the authors cover range size. Results, therefore, are valid by assuming small population size (abundance) and high habitat specialization for all species across their range, and that the traits being used can capture variation in abundance and habitat specialization. These assumptions may not always hold because a small-ranged species can be locally very abundant (especially in islands), can be a generalist when choosing preferred habitats, and can have traits regionally common and not linked with any demographical process.

As a reader, I would like to see all rarity facets being covered here. First, for abundance, perhaps you could use the probability of occupancy (as already used by authors to define rare or common species, line 143 of your manuscript) as a proxy for abundance, as high occupancy probability generally is correlated with large abundance. Second, I am almost sure you can extract some sort of information about "habitat specialization" from FishBase (even though I think that FishBase should never be used blindly – e.g. Robertson 2008 - Global biogeographical databases on marine fishes: caveat emptor -- <https://onlinelibrary.wiley.com/doi/full/10.1111/j.1472-4642.2008.00519.x>). Finally, I would take care about the traits; I would use life-story traits linked to demographic processes [which might be influenced by the surrounding environment] generating ecological rarity. Among the traits you already used, I would go with "growth", "generation time",

"length of first maturity", "reproductive guild"; depending on the correlation with other traits I would include "body size".

Most of the values of geographical restrictedness GE [calculated as $GE = 1 - (\text{number of occupied cells} / \text{total number of cells})$] are very close to one. These high values are caused by the very large total number of cells relative to the number of occupied cells (e.g., for a species with 2 cells in a universe of 1000 cells, the GE would be equal to $1 - (2 / 1000) = 0.998$). Restricting the total number of cells by oceanic region could help to at least decrease the total number of cells. However, I think the main problem is explained in Line 151: "In our analysis, we selected only occurrence data (the presence of a given species in a certain half-degree grid cell) with a probability higher than 0.9". This means that only a tiny part of the species range is included in these maps. In addition, rare species, by definition, have a low abundance and likely low probabilities of occurrence in most cells. Hence authors should consider using maps with probability thresholds lower than 0.9. To avoid subjective decisions about probability thresholds, I would calculate GE as $GE = 1 - (\text{sum of probabilities of species } i / \text{total number of cells in region } r)$.

A 5-degree cell size can be too coarse for both define rarity and test the mismatch between rarity and location of MPAs. For instance, many of these grid cells cover land along with the coastal sites, which will likely influence how much rarity (%) is within marine protected areas. Why using this resolution if original data have a resolution of 0.5°? Authors could do a better job here.

While the number of traits might not influence the results, the functional index (either functional distinctiveness or uniqueness) being used had a large influence on results. More rarity can be seen by using uniqueness rather than distinctiveness. Authors should consider showing results for both indexes (and discuss these findings) as they provide complementary information about functional rarity.

Functional distinctiveness/uniqueness were calculated based on a global rather than a regional pool of species. In the present calculation, one fish from e.g., Australia can be compared to a fish from e.g., UK. Fishes from different regions likely will have different traits because they have quite different evolutionary histories. I would include in my regional pool the species found in each Oceanic region (as shown in Extended Data Table 1).

Sometimes it was difficult to understand when authors used only geographic restrictedness, only functional distinctiveness, or both. This is in part because they did not show a map for functional distinctiveness in Fig. 1. Furthermore, it is not easy to understand whether authors used either raw or residual rarity when defined the top rarity values. These points need more clarification.

Spatial autocorrelation can be an important issue here, mainly because authors made inferences on environmental drivers based on adjacent grid cells. Generalized Least Squares regression or Geographically Weighted Regression can do the job.

MS “Biogeography of rare fish in the world’s oceans”. The authors did a great job in gathering big datasets to address spatial patterns of rarity across the oceans, and checking the superimposition between rarity and the location of marine protected areas (MPAs). While I believe this manuscript is of interest for the audience of Nature Communications, I identified a set of issues that prevent me from accepting this manuscript in its present form.

Rabinowitz’s concept of rarity embraces three facets: range size, abundance, habitat specialization. In this study, authors cover range size. Results therefore are valid by assuming small population size (abundance) and high habitat specialization for all species across their range, and that the traits being used can capture variation in abundance and habitat specialization. These assumptions may not always hold because a small-ranged species can be locally very abundant (especially in islands), can be generalist when choosing preferred habitats, and can have trait regionally common and not linked with any demographical process.

As a reader, I would like to see all rarity facets being covered here. First, for abundance, perhaps you could use the probability of occupancy (as already used by authors to define rare or common species, line 143 of your manuscript) as a proxy for abundance, as high occupancy probability generally is correlated with large abundance. Second, I am almost sure you can extract some sort of information about “habitat specialization” from FishBase (even though I think that FishBase should never be used blindly – e.g. Robertson 2008 - Global biogeographical data bases on marine fishes: caveat emptor -- <https://onlinelibrary.wiley.com/doi/full/10.1111/j.1472-4642.2008.00519.x>). Finally, I would take care about the traits; I would use life-story traits linked to demographic processes [which might be influenced by the surrounding environment] generating ecological rarity. Among the traits you already used, I would go with “growth”, “generation time”, “length of first maturity”, “reproductive guild”; depending on the correlation with other traits I would include “body size”.

Most of values of geographical restrictedness GE [calculated as $GE = 1 - (\text{number of occupied cells} / \text{total number of cells})$] are very close to one. These high values are caused by the very large total number of cells relative to the number of occupied cells (e.g., for a species with 2 cells in a universe of 1000 cells, the GE would be equal to $1 - (2 / 1000) = 0.998$). Restricting the total number of cells by oceanic region could help to at least decrease the total number of cells. However, I think the main problem is explained in Line 151: "In our analysis we selected only occurrence data (the presence of a given species in a certain half degree grid cell) with a probability higher than 0.9". This means that only a tiny part of species range is included in these maps. In addition, rare species, by definition, have low abundance and likely low probabilities of occurrence in most cells. Hence authors should consider using maps with probability thresholds lower than 0.9. To avoid subjective decisions about probability thresholds, I would calculate GE as $GE = 1 - (\text{sum of probabilities of species } i / \text{total number of cells in region } r)$.

A 5-degree cell size can be too coarse for both define rarity and test the mismatch between rarity and location of MPAs. For instance, many of these grid cells cover land along the coastal sites, which will likely influence on how much rarity (%) is within marine protected areas. Why using this resolution if original data have a resolution of 0.5°? Authors could do a better job here.

While the number of traits might not influence the results, the functional index (either functional distinctiveness or uniqueness) being used had a large influence on results. More rarity can be seen by using uniqueness rather than distinctiveness. Authors should consider to show results for both indexes (and discuss these findings) as they provide complementary information about functional rarity.

Functional distinctiveness/uniqueness were calculated based on a global rather than a regional pool of species. In the present calculation, one fish from e.g., Australia can be compared to a fish from e.g., UK. Fishes from different regions likely will have different traits because they have quite different evolutionary histories. I would include in my regional pool the species found in each Oceanic region (as shown in Extended Data Table 1).

Sometimes it was difficult to understand when authors used only geographic restrictedness, only functional distinctiveness, or both. This is in part because they did not show a map for functional distinctiveness in Fig. 1. Furthermore, it is not easy to understand whether authors used either raw or residual rarity when defined the top rarity values. These points need more clarification.

Spatial autocorrelation can be an important issue here, mainly because authors made inference on environmental drivers based on adjacent grid cells.

Generalized Least Squares regression or Geographically Weighted Regression can do the job.

Other comments are presented below (in blue).

Specific comments

Line 47: Hypothesis: “Based on previous findings that the tropics are the hotspots for marine species richness (taxonomic diversity) but have high levels of functional redundancy, e.g. species playing similar ecological roles 25,26 we expect to find hotspots of functional rarity towards higher latitudes.”

Why not: “Based on previous findings that the tropics are the hotspots for marine species richness (taxonomic diversity) but have high levels of functional redundancy, e.g. species playing similar ecological roles 25,26 we expect to find low functional rarity in the tropics.”

Line 45: “We define rare taxa as those that occur in the top (i.e. most rare) quartiles of both distributions”

Which “distributions”? You did not describe any distribution of values in the previous sentences.

Line 39: I don't like "Taxonomic rarity". It reminds me e.g., phylogenetic uniqueness, evolutionary distinctiveness. I would always call "geographic restrictedness" to avoid confusion.

Line 59 - "when we control for species richness by taking the residuals of the relationship between rarity and species richness, our finding that coastal regions are associated with high levels of rarity is reinforced".

The problem of using residuals, as shown by Nathan Swenson in his book about phylogenetic and functional analysis (2011), is that the variance of functional indexes is still correlated with species richness. This occurs because you can find both high and low values of rarity at low levels of richness, but you likely will find an average (and more precise estimate of) rarity at high levels of richness. In that book you will find constrained null models that you should consider to use here.

Line 79: "we compared these results (Extended Data Fig. 4) with an alternative method of calculating rarity (using functional uniqueness (Extended Data Fig. 2) rather than functional distinctiveness, see Extended Data Fig. 5 to 7) to ensure the robustness of our approach; the conclusions were consistent (see Extended Data Fig. 6 and 7)."

I could not find support for these statements in the results. First, Extended Data Fig. 2 shows a correlation very close to zero between distinctiveness and uniqueness for the two groups of fishes. This means that distinctiveness and uniqueness are complementary sources of information about functional rarity. Second, we can see much more areas with rarity in Extended Data Fig. 6 than in Fig. 1. Therefore, results were sensitive to the functional index being used.

Fig. 1, Functional distinctiveness

Extended Data Fig. 6, Functional uniqueness

Line 83: "It is possible that a sampling bias towards records in coastal waters could partly explain our findings. However, we note that regions of the earth that are typically under-recorded in biodiversity surveys, such as South America and Africa, are identified in our analysis as important repositories of rarity, while marine areas that have been intensively monitored, such as the coasts of NW Europe are not."

This might be an artifact of using rarity ~ richness residuals, as mentioned above. I would consider the use of null models.

Line 106: Why discussing patterns of biodiversity in land?

Fig. 1: it is hard to see the contrast between cells, as well fish silhouettes (mainly in 'd'). Why not use black silhouettes?

Extended Data Fig. 1 – not enough resolution to see numbers and formulae.

Line 192: “Extended Data Fig. 3 c and d, these traits are largely uncorrelated.”

I think this figure means exactly the opposite. If the relation is maintained if you remove 1 trait, then the information removed is still in the remaining traits.

Line 542: “Functional distinctiveness is measured by the average of the functional distance of a given species compared with all other species in the regional pool”.

Does it mean that, for example, a species from Australia was compared to a species from UK? I would impose more restrictions to this “regional pool” (which is truly the global pool of fish species). Perhaps you should restrict Functional distinctiveness calculations within the pool of biogeographic (e.g., oceanic) regions (like you showed in Extended Data Table 1).

Line 559: Extended Data Fig. 3.

It is lacking Y-axis labels. Would it be frequency?

Line 562: Extended Data Fig. 2. Distribution of correlations between trait values for the Actinopterygii (c) and Elasmobranchii (d). Correlations are weak and largely centered on zero, particularly for the Actinopterygii, indicating that traits provide complementary information on function”

It means that uniqueness and distinctiveness provide complementary information about rarity. As there is no correlation at all there's no reason to show the results for only one functional index.

Line 567: Extended Data Fig. 4 – The distribution of restrictedness is very skewed, with a large number of species presenting high restrictedness; that's why your horizontal red line is very close to 1. That's because you have a very large total number of cells relative to number of cells occupied by rare species in the cells. I would consider other ways to calculate GE (see comments in the beginning of this letter).

Line 581: Extended Data Fig. 6 – one silhouette is superimposing “residuals” in d

Supplementary table 2: please rank your models by increasing AIC values (or instead by AICw, which you must show as it depicts the weight of one model relative to the other models).

This letter is the response to the reviewers' comments:

Dear anonymous reviewers,

We thank for this opportunity to address the high-level points raised. We have now performed the additional requested analyses and are pleased to report that they are consistent with the results and conclusion we reported in our first submission.

First, we would like to respond to the specific main concerns raised. We hope that this will provide an overall overview of all changes and findings after following the suggestions. Then we will address each point by point comments made by both reviewers.

Main points raised.

Point one:

- The reviewers are concerned that the way functionally rarity was computed may have led to confounding functionally rare species *sensu stricto* with species whose rarity is caused by local features.

Answer: The referees raised this important issue due to concerns that local features, such as temperature and depth, could explain our findings. To address this question we carried a sensitivity analysis to test if the traits “mean temperature preference” and “maximum depth” influenced our results. As we can see in Extended Data Fig. 8, the sensitivity analysis shows that even when we remove those traits from the analysis the results are consistent. This supports our argument that the global patterns of rarity we report are not simply a response of traits influenced by local features (more details can be found at the section *Further Analysis* at the methods section from the paper (lines between 301 and 310).

Point two:

- The reviewers were concerned that using residuals of the richness-rarity relationship is not an appropriate way to control for confounding effect of species richness; Reviewer 2 recommends using null models instead.

In our revised analysis we built a null model (following the curveball algorithm developed by Strona, et al. ¹) and ran this for 2000 iterations to compare the observed number of rare species in the grid cells with the null expectation (as defined by Standardized Effect Sizes (SES)). In brief the method was as follows:

$$SES = (X - Y)/Z$$

X as the number of rare species observed in each grid cell,

Y as the average of rare species found from the null model after 2000 interactions and

Z as the standard deviation from Y.

A positive SES indicates more rare species than would be expected by chance and a negative SES fewer than expected.

The map in Figure 1 shows the biogeographic distribution of raw rarity for bony fish and cartilaginous fish (Figure 1a and c respectively) in Coastal Systems. The grid cells with concentrations of rare species higher than expected by chance are in red and lower than expected in blue (Figure 1b and d). These results are also in concordance with our previous findings (see *Null Model* section at the methods for more details, lines between 259 and 285). We also include the results for the High Seas in the main text (Figure 2).

Point three:

- Inclusion of both coastal and pelagic species in the analysis may need to be reconsidered.

As suggested, we performed our analyses for each of the seven Oceanic Regions individually (those are: North Atlantic, South Atlantic, Indian Ocean, North West Pacific, South West Pacific, North East Pacific and South East Pacific (Extended Data Fig. 3)). We ran these analyses separately for the Coastal Systems and High Seas from those seven Oceanic Regions. As in the previous analyses, the species were split by bony fishes and cartilaginous fishes.

See lines between 279 and 285:

“We then have the following “roadmap”:

- i. Scales – 7 Coastal Systems and 7 High Seas.
- ii. Groups – bony and cartilaginous fish.
- iii. Indices – distinctiveness and uniqueness.
- iv. Probability of occurrences – > 0.9 , > 0.7 and > 0.5 .
- v. Total – 168 **independent cases** analysis (each having its own assemblage and trait matrices.”

These new results support our previous conclusion that there are higher concentrations of rare species towards higher latitudes and near the continents. As noted above, in this analysis the Seven Coastal Systems and Seven High Seas were investigated separately.

- The analysis should include proxies for abundance and habitat specialization (Reviewer 2’s specific suggestions).

To address the first concern raised by the reviewer, we conducted our analysis using 3 different probabilities of occurrence. Those were: probability > 0.9 , probability > 0.7 and probability > 0.5 . AquaMaps² gives probabilities of the occurrence of a given species between 0 (based on the environmental envelope there is no chance of finding the species in that grid cell) to 1 (highest probability to find the species in that grid cell). We therefore conducted all the analyses and performed the null model with 2000 runs for each of those 28 independent case systems using each of those 3 different probabilities of occurrence. These new results support our earlier findings. In all cases, we find an excess of rare species in higher latitudes (we now provide the results for each probability as follows:

Figure 3 = probability > 0.9 .

Extended Data Fig. 6 = probability > 0.7 and > 0.5 .

With regard to the second concern about habitat specialization, we are pleased to include this analysis in our work (see *Habitat Specialization* section at the Methods section (between lines 341 and 348)). At the updated version of the methods, we have:

“*Habitat Specialization*

All species are classified according to its position in water column (bathymersal, bathypelagic, benthopelagic, demersal, pelagic neritic, pelagic oceanic and reef associated (as provided in FishBase³)) and here we are using this trait as a proxy for “habitat specialization”. We then used a G test, and Cramer’s V (using the functions *GTest* and *CramerV* from the package *DescTools*⁴) to compare the

frequency distribution in habitat specialization between rare and non-rare species.” (between lines 341 and 348).

At the updated version of the main text, we then have:

“Do these patterns of rarity reflect heterogeneity in habitat use by the rare taxa? To explore this we compared the frequencies of “habitat specialization” categories (i.e. bathydemersal, bathypelagic, benthopelagic, demersal, pelagic neritic, pelagic oceanic and reef associated) in the rare and non-rare species and found evidence of differences in their association with different habitat types (Extended Data Fig. 9). For example, in the Northwest Pacific Coast most non-rare species are classified as reef associated (Extended Data Fig. 9 c), while rare species are mostly classified as demersal (Extended Data Fig. 9 a).” (lines between 103 and 109).

- A finer spatial resolution than the current 5-degree cell size should be used.

These new analyses were done using 2-degree grid cells; this increased considerably the number of grid cells in each of the systems investigated (see Extended Data Fig. 3).

- Spatial autocorrelation would have to be accounted for.

We constructed distance decay plots to examine spatial autocorrelation and fitted a quantile regression to these relationships. The results are illustrated in Extended Data Fig. 7 which shows the distance decay calculated by pairwise differences (Extended Data Fig. 7 a – Coastal Systems and b – High Seas for bony fish, and Extended Data Fig. 7 c – Coastal Systems and d – High Seas for cartilaginous fish) between a given grid cell and all other grid cells present in the Northwest Pacific Ocean. These plots provide reassurance that spatial autocorrelation is not obscuring the results we report. (see section *Spatial autocorrelation* between lines 311 and 318).

Now answering point by point the reviewers suggestions and comments:

Reviewer #1 (Remarks to the Author):

Dear Editor,

Thank you very much for giving me the opportunity to review the paper "Biogeography of rare fish in the world's oceans". The Authors highlight how current estimates of rarity focus on species' geographic ranges. As an alternative, they propose to measure rarity by looking not only at species range but also at species functional distinctiveness and uniqueness (where rare species have a trait combination that does not occur in other species). After obtaining such a measure for a large set of fish species (by using information on various life history and ecological traits obtained from FishBase), they reprojected rarity at the global scale (at a relatively coarse resolution of 5 x 5 degrees), as the number of rare species identified according to their new criterion. To avoid obvious potential biases due to the expected relationship between overall local species richness and rare species' richness, they quantified local rarity by looking at the residuals of the overall rarity/richness relationship. They then explored geographical patterns of rarity, the overlap/mismatch between rarity hotspots and protected areas, and the effect of environmental variables on rarity. They also performed a sensitivity analysis to explore the impact of functional traits' choice on the rarity patterns.

They found that rarity is higher in coastal areas and higher latitudes, with various (quite large) areas of hotspots. Although not coincident, the pattern is relatively consistent for bony fish and cartilaginous fish. The Authors also found a widespread mismatch between rarity hotspots and protected areas.

I think the study is interesting and has value, but I have a few general conceptual and methodological concerns which make me cautious about the results and their interpretation.

1) Goal/novelty of the study

Reading the abstract and the Introduction, a reader has the impression that the Authors have developed a new concept of rarity going beyond species range (i) and then they have applied that for the first time to fish (ii) at the global scale (iii). Of these three (implicit?) claims, only the 3rd one is true. The idea of considering functional traits to estimate rarity is actually not new and very well discussed, for example, in this perspective paper: <https://doi.org/10.1016/j.tree.2017.02.002>. The approach has been already applied to fish, with a very similar conceptual and analytical workflow: <https://doi.org/10.1016/j.biocon.2018.08.011>

That paper examines functional (and evolutionary) rarity patterns for more than 2000 reef fish globally. So the main novelty here is the application of the approach at a global scale not limited to the tropics. This should be clarified in the Introduction. A brief discussion of the previous studies/results would also be helpful to set up the scene. The title is quite generic and not informative about the paper's goals/findings.

Answer:

We thank the reviewer for raising all those important points and suggestions. We now make it clearer in the introduction that the novelty of our work is based on the global scale adopted in our study and, more importantly, by the biogeographical patterns we report. Following the suggestions made by the reviewers and the editor we now use Seven Oceanic Regions subdivided into coastal and high seas (7 of each, see lines between 54 and 58, see Extended Data Fig. 3). All our analysis were done separately for those 14 different systems (as mentioned above). We also made it clearer that we defined rarity based on the conceptualization created by Rabinowitz⁵ (for geographic restrictedness) and Grenié⁶ (for global distinctiveness) (see lines between 47 and 50). We also applied a finer grid cell scale in our

work; all maps were constructed using 2° grid cells (see Occupancy Data section between lines 163 and 184).

2) Scale of functional rarity and choice of the functional/ecological traits

I am not 100% I understood this correctly, but my impression is that the Authors assessed rarity by looking at the complete set of fish species (separately for bony and cartilaginous fish). I suspect that this, combined with the choice of the environmental traits (especially maximum depth, mean temperature preference) and some adaptive traits (such as swimming mode, body shape), might create some circularity. That is, the measure of global rarity for a given species might partly reflect the availability of habitat/conditions suitable to the species. Then, I might expect to have more niches for rare species where the local environmental setting is rarer.

To make an extreme example, I can imagine that a specific set of environmental/ecological conditions can be found only in a few localities worldwide. Some species might be well adapted to those particular conditions. Therefore, their range would be likely restricted to that locality, and their functional traits would also reflect the peculiarity of the environment. The Authors' approach would therefore identify those species as rare and the area where the species occur as a hotspot of rarity. This is fine, but I think that it would be quite important to discriminate between this kind of situations (where local rarity is a consequence of local environmental/ecological features) and a situation where rarity depends only on species. From a conservation perspective, this is very important (as the first case would call for management strategies focusing on the site; while the latter would require strategies focusing on the species).

The confounding effect of the role of the environment on species traits (and hence the potential link between global patterns of environmental variability and global patterns of rarity) could be possibly reflected in the large scale overall results. Especially, the main findings of more rarity in the coastal areas and at higher latitudes seem to me as potential consequences of this mechanism. Coastal environments clearly provide more environmental heterogeneity (and hence possibly more specialized habitats promoting the presence of rare species according to the Authors' metric) compared to the pelagic realm. The pattern of increasing rarity with latitude might be even more an effect of the analytical approach: by pooling all species together when calculating rarity, given the higher fish diversity of the tropics and the selectivity of the high-latitude environments, it is not surprising to find more rare species in temperate and cold-waters realms than in the tropics (which in turn drives the observed patterns). The Authors tried to control for this effect, but see my comments below on their approach.

Answer:

Thank you for highlighting these issues. Following your suggestions, we did all the analysis separately for all Coastal Systems (Figure 1) and High Seas systems (Figure 2). In addition, yes, we kept bony and cartilaginous fish separate. Furthermore, we also tested the possible influence of the “environmental traits” (such as depth and mean temperature preference) in our findings. To do this, we performed a sensitivity analysis specifically removing each of those traits in turn (as well as both traits together) and then we compared the species distinctiveness and uniqueness found without those traits with all 11 traits together. Comparing both patterns, we found a strong correlation analysis between all traits (x-axis) and all traits without the “environmental traits” (y-axis) (see Extended Data Fig. 8). Therefore, we show here that the results found cannot be explained as an artefact of the effect of those “environmental traits”. We believe that doing this test improved the robustness of our study and thank reviewers and the editor for the suggestion.

One potential solution to the issue is that of examining patterns of rarity by splitting the global species pool into subsets based on some biogeographical/environmental criteria. The issue was considered

also in the aforementioned paper about reef fish functional rarity (“We tested the influence of the species pool definition from which functional distinctiveness was extracted. To this aim, we split the cells into two realms—Atlantic and Pacific—and estimated species distinctiveness values separately in each realm”).

Answer:

As we explained above, in the revision we ran the analysis for each of the 14 different systems independently, and for bony and cartilaginous fish separately as well.

Selecting a different set of functional traits not reflecting adaptation to specific habitat conditions could be another potential strategy, but I am not sure which traits I would use. Perhaps removing at least the mean temperature preference and/or the max depth could be one starting point, but I guess the issue is a bit philosophical as every single trait could be seen as an adaptation to some environment and I cannot see any strong rule or argument to identify traits that are more or less important to define a species rarity.

Answer:

As explained at the beginning of the letter, we followed this suggestion and showed that this is not the case.

To this end, it would also be important to consider that the concept of rarity is a very relative one (while the Authors seem to present it as a quite absolute measure). In this regard, I think Table 1 in the perspective paper mentioned above (<https://doi.org/10.1016/j.tree.2017.02.002>) provides a valuable roadmap to the different possible types (12) of functional rarity. I am not suggesting that the Authors need to explore all the possible types of rarity, but at least they should clarify where their measure falls into the classification scheme.

3) Disentangling the effect of richness from rarity.

Considering the potential issues of circularity highlighted above, I appreciate that the Authors attempted to correct for the potential effect of species richness. However, I have a few concerns about their approach (looking at the residuals of the linear regression of richness vs. rarity). I could not find in the text or SI information on the actual regression (apologies if I missed it). Is the relationship really linear and well fitted by the regression? Otherwise looking at the residuals would not be very reliable. Grenie’ et al. (in the global reef fish rarity paper referred to above, <https://doi.org/10.1016/j.biocon.2018.08.011>) used an explicit null model approach that seems much more flexible and robust to me (they compared the observed patterns with a set of randomized species/locality matrices obtained by reshuffling species occurrences while keeping fixed the number of occurrences per locality, and the number of localities where each species occurs; they assessed local rarity in terms of standard effect size from the comparison of observed vs. null rarity).

In addition, linking this aspect back with the circularity issue above, I think that the approach of looking at the residuals of the richness-rarity relationships might even worsen the issue. In fact, my intuition is that measuring richness on the global species pool would tend to identify rare species in the areas with lower diversity (because of the more selective environment). However, due to the large global variation in species richness (with orders of magnitude differences in the number of species per 5x5 grid cell globally), this effect would tend to disappear/be reversed in the overall relationship between richness and rarity (which will be, I think/infer from the text, a positive relationship). Consequently, looking at the residuals will reinforce the expectation of rarity in areas with low diversity (looking at the issue from another perspective, I think using the OLS residuals is

conceptually close to showing relative rarity, i.e. the fraction of rare species per total local richness; the impact of a rare species will be much higher in a locality with low diversity compared to a hotspot of diversity, which might explain the results in fig. 1 (with the usual hotspots of diversity showing very low rarity).

Answer:

Thank you for encouraging us to explore the accuracy of our findings by applying a null model. We proceeded as follows (all details now are included in the new version of our paper (see lines between 259 and 289)):

“Step 5 – Null Model

Does the number of rare species in a given grid cell differ from the null expectation? To answer this question we applied a null model approach based on the curveball algorithm 68. This algorithm keeps constant the total number of species (rare + non-rare) and the number of grid cells that each species occurs. It then randomizes the presence and absence of all species following these thresholds. We ran the model for 2000 iterations; in each loop it randomizes the occurrences of all species, identifies where the rare species are falling and then counts the total number of rare species in each grid cell.

To quantify how the observed number of rare species differ from the null expectation we then use Standardized Effect Sizes (SES) as follows:

$$SES = (X - Y)/Z$$

X as the number of rare species observed in each grid cell,

Y as the average of rare species found from the null model after 2000 interactions and

Z as the standard deviation from Y.

A positive SES indicates more rare species than would be expected by chance and a negative SES fewer than expected.

Since we are using 14 different systems (7 Coastal Systems and 7 High Seas (see Extended Data Fig. 3)) for 2 groups of organisms (bony and cartilaginous fish), 2 functional rarity indices (distinctiveness and uniqueness), and using 3 different probabilities of occurrences (prob. > 0.9, > 0.7 and > 0.5).

We then have the following “roadmap”:

- i. Scales – 7 Coastal Systems and 7 High Seas.
- ii. Groups – bony and cartilaginous fish.
- iii. Indices – distinctiveness and uniqueness.
- iv. Probability of occurrences – > 0.9, > 0.7 and > 0.5.
- v. Total – 168 independent cases analysis (each having its own assemblage and trait matrices).

Therefore, as mentioned above, the null model ran 2000 interactions for each of those independent cases. The final matrices from these initial steps contain grid cells as rows and as columns we have the raw number of rare species along with the SES values for each. These matrices were important to map our results.” (see lines between 260 and 289).

The Figure 1 a and c respectively show the biogeographic distribution of raw rarity for bony fish and cartilaginous fish in Coastal Systems. The grid cells with concentrations of rare species higher than expected by chance are in red and lower than expected in blue (Figure 1b and d). These results are

also in concordance with our previous findings. Figure 2 will show the results found for the High Seas (Figures 1 and 2 were done using Restrictedness and Distinctiveness as functional index), meanwhile the Extended Data Fig. 4 and 5 will show the results found by using the Restrictedness and Uniqueness as the functional index.

(4) Coastal vs. pelagic

The low (almost none?) rarity in the pelagic realm might be the result of two aspects. The first one is the fact that most pelagic species have wide ranges (and the same issue applies in part to the results for sharks) and, by necessity, quite generalistic traits (not surprising in a heterogeneous environment, see point 2). Second, there is an obvious sampling bias when comparing pelagic vs. coastal habitats. The very limited knowledge about deep fish is clearly preventing us from making a fair comparison of rarity, as we cannot tell how many rare species are there at the ocean bottoms. I am not saying that the Authors should fill in this knowledge gap, I am just saying that treating coastal habitats in the same way as pelagic habitats is a bit unbalanced, and I would have probably not done that.

Answer:

As suggested, we performed our analyses for each of the seven Oceanic Regions individually (those are: North Atlantic, South Atlantic, Indian Ocean, North West Pacific, South West Pacific, North East Pacific and South East Pacific (Extended Data Fig. 3)). We also ran these analyses separately for the Coastal Systems (Figure 1) and High Seas (Figure 2) from those seven Oceanic Regions. As in the previous analyses, the species were split by bony fishes and cartilaginous fishes. These new results support our previous conclusion that there are higher concentrations of rare species towards higher latitudes and near the continents. In this analysis the Seven Coastal Systems and Seven High Seas were investigated separately.

(5) Ranges vs. abundance

As previously shown (<https://www.nature.com/articles/nature12529>), accounting for species abundances is fundamental to explore patterns of diversity and rarity. Again, I am not saying that the Authors should do this here, but I think that it would quite important to at least acknowledge this (see also the importance of population abundance in the definitions of rarity in the above-mentioned Table 1 from Violle et al.). In particular, it is true that many pelagic species are still widespread, but the populations of many of those are dramatically declining. Same for sharks.

Answer:

Unfortunately, due to the lack of abundance data at global scale for all described marine fish species, we could not include this information in our study. However, we explored an alternative approach to reinforce the robustness of our findings. To address the concern raised by the reviewers and the editor, we conducted our analysis using 3 different probabilities of occurrence.

Those were: probability > 0.9, probability > 0.7, probability > 0.5 (see *Occupancy Data (assemblage matrix)* section for further details, between lines 163 and 184).

AquaMaps² gives probabilities of the occurrence of a given species between 0 (based on the environmental envelope there is no chance of finding the species in that grid cell) to 1 (highest probability to find the species in that grid cell). We therefore conducted all the analyses and performed the null model with 2000 runs for each of those 28 independent case systems using each of those 3 different probabilities of occurrence (see *Occupancy Data (assemblage matrix)* section at the Methods, between lines 163 and 184).

Reviewer #3 (Remarks to the Author):

MS “Biogeography of rare fish in the world’s oceans”. The authors did a great job in gathering big datasets to address spatial patterns of rarity across the oceans and checking the superimposition between rarity and the location of marine protected areas (MPAs). While I believe this manuscript is of interest to the audience of Nature Communications, I identified a set of issues that prevent me from accepting this manuscript in its present form.

Rabinowitz’s concept of rarity embraces three facets: range size, abundance, habitat specialization. In this study, the authors cover range size. Results, therefore, are valid by assuming small population size (abundance) and high habitat specialization for all species across their range, and that the traits being used can capture variation in abundance and habitat specialization. These assumptions may not always hold because a small-ranged species can be locally very abundant (especially in islands), can be a generalist when choosing preferred habitats, and can have traits regionally common and not linked with any demographical process.

As a reader, I would like to see all rarity facets being covered here. First, for abundance, perhaps you could use the probability of occupancy (as already used by authors to define rare or common species, line 143 of your manuscript) as a proxy for abundance, as high occupancy probability generally is correlated with large abundance. Second, I am almost sure you can extract some sort of information about “habitat specialization” from FishBase (even though I think that FishBase should never be used blindly – e.g. Robertson 2008 - Global biogeographical databases on marine fishes: caveat emptor - <https://onlinelibrary.wiley.com/doi/full/10.1111/j.1472-4642.2008.00519.x>). Finally, I would take care about the traits; I would use life-story traits linked to demographic processes [which might be influenced by the surrounding environment] generating ecological rarity. Among the traits you already used, I would go with “growth”, “generation time”, “length of first maturity”, “reproductive guild”; depending on the correlation with other traits I would include “body size”.

Answer:

We are pleased to incorporate these suggestions in our revision. Specifically we include: (i) a new sensitivity analysis showing that traits related with the environment were not driving the patterns we found (see Extended Data Fig. 8), (ii) using a smaller grid cell size (see *Occupancy Data (assemblage matrix)* section at the Methods, between lines 163 and 184), (iii) splitting the analysis across Coastal Systems and High Seas (see Extended Data Fig. 3), (iv) the habitat specialization analysis, where we show that rare and non-rare species are emerging with different habitat specializations (one is not the reflection of the other (Extended Data Fig. 9)). We describe all those changes earlier in the response letter. We are grateful for the advice of the anonymous reviewers.

Most of the values of geographical restrictedness GE [calculated as $GE = 1 - (\text{number of occupied cells} / \text{total number of cells})$] are very close to one. These high values are caused by the very large total number of cells relative to the number of occupied cells (e.g., for a species with 2 cells in a universe of 1000 cells, the GE would be equal to $1 - (2 / 1000) = 0.998$). Restricting the total number of cells by oceanic region could help to at least decrease the total number of cells. However, I think the main problem is explained in Line 151: “In our analysis, we selected only occurrence data (the presence of a given species in a certain half-degree grid cell) with a probability higher than 0.9”. This means that only a tiny part of the species range is included in these maps. In addition, rare species, by definition, have a low abundance and likely low probabilities of occurrence in most cells. Hence authors should consider using maps with probability thresholds lower than 0.9. To avoid subjective decisions about probability thresholds, I would calculate GE as $GE = 1 - (\text{sum of probabilities of species } i / \text{total number of cells in region } r)$.

Answer:

Regardless of the probability of occurrences selected we find consistent results, with greater concentrations of rare species towards higher latitudes than expected by chance based on the null expectation.

A 5-degree cell size can be too coarse for both define rarity and test the mismatch between rarity and location of MPAs. For instance, many of these grid cells cover land along with the coastal sites, which will likely influence how much rarity (%) is within marine protected areas. Why using this resolution if original data have a resolution of 0.5°? Authors could do a better job here.

Answer:

Thank you for your suggestion; all the analyses now use 2° grid cells (Extended Data Fig. 3).

While the number of traits might not influence the results, the functional index (either functional distinctiveness or uniqueness) being used had a large influence on results. More rarity can be seen by using uniqueness rather than distinctiveness. Authors should consider showing results for both indexes (and discuss these findings) as they provide complementary information about functional rarity.

Answer:

We present the results for uniqueness in the Extended Data Fig. 4 and 5 for Coastal and High Seas respectively in our updated paper version.

Functional distinctiveness/uniqueness were calculated based on a global rather than a regional pool of species. In the present calculation, one fish from e.g., Australia can be compared to a fish from e.g., UK. Fishes from different regions likely will have different traits because they have quite different evolutionary histories. I would include in my regional pool the species found in each Oceanic region (as shown in Extended Data Table 1).

Answer:

As we provided detailed information above, we also followed this specific suggestion. Thank you!

Sometimes it was difficult to understand when authors used only geographic restrictedness, only functional distinctiveness, or both. This is in part because they did not show a map for functional distinctiveness in Fig. 1. Furthermore, it is not easy to understand whether authors used either raw or residual rarity when defined the top rarity values. These points need more clarification.

Answer:

Both figures: (i) Figure 1 a and c (for Coastal Systems), and (ii) Figure 2 a and c (for High Seas) show raw rarity (that is, the number of rare species in each grid cell).

Meanwhile Figure 1 b and d, Figure 2 b and d, both represents the SES values. We hope this is clearer. Thank you.

(The same for Extended Data Fig. 4 and 5).

Spatial autocorrelation can be an important issue here, mainly because authors made inferences on environmental drivers based on adjacent grid cells. Generalized Least Squares regression or Geographically Weighted Regression can do the job.

Answer:

We have addressed the autocorrelation concern as addressed above.

Other comments are presented in the supplementary file

Specific comments Line 47: Hypothesis: “Based on previous findings that the tropics are the hotspots for marine species richness (taxonomic diversity) but have high levels of functional redundancy, e.g. species playing similar ecological roles 25,26 we expect to find hotspots of functional rarity towards higher latitudes.”

Why not: “Based on previous findings that the tropics are the hotspots for marine species richness (taxonomic diversity) but have high levels of functional redundancy, e.g. species playing similar ecological roles 25,26 we expect to find low functional rarity in the tropics.”

Answer:

Thank you. Done (see lines 50 and 53).

Line 45: “We define rare taxa as those that occur in the top (i.e. most rare) quartiles of both distributions”

Which “distributions”? You did not describe any distribution of values in the previous sentences.

Answer:

We hope that now it is clearer. As follows:

“We define rare taxa as those that occur in the top (i.e. most rare) quartiles ⁷ of both: restrictedness and distinctiveness. Our measure of rarity thus reflects both taxonomic rarity (geographic restrictedness) and functional rarity (global functional distinctiveness and functional uniqueness), i.e. facets of rarity conceptualized respectively by Rabinowitz ⁵ and Grenié, et al. ⁶.” (see lines between 46 and 50).

Our intention was to refer to the values distribution of both indices across all species.

Line 39: I don't like “Taxonomic rarity”. It reminds me e.g., phylogenetic uniqueness, evolutionary distinctiveness. I would always call “geographic restrictedness” to avoid confusion.

Answer:

Thank you. We now reworded as follows:

“...in order to calculate a **key facet of rarity, namely species geographic restrictedness** ⁵ (see Extended Data Figs.1 for workflow and Extended Data Figs. 2 for geographic restrictedness calculation).” (lines between 39 and 41).

Line 59 - “when we control for species richness by taking the residuals of the relationship between rarity and species richness, our finding that coastal regions are associated with high levels of rarity is reinforced”.

The problem of using residuals, as shown by Nathan Swenson in his book about phylogenetic and functional analysis (2011), is that the variance of functional indexes is still correlated with species richness. This occurs because you can find both high and low values of rarity at low levels of richness,

but you likely will find an average (and more precise estimate of) rarity at high levels of richness. In that book you will find constrained null models that you should consider to use here.

Answer:

Thank you for allowing us to explore the accuracy of our findings by applying a null model. We proceeded as follows (all details now are included in the new version of our paper (see lines between 259 and 289)):

“Step 5 – Null Model

Does the number of rare species in a given grid cell differ from the null expectation? To answer this question we applied a null model approach based on the curveball algorithm 68. This algorithm keeps constant the total number of species (rare + non-rare) and the number of grid cells that each species occurs. It then randomizes the presence and absence of all species following these thresholds. We ran the model for 2000 iterations; in each loop it randomizes the occurrences of all species, identifies where the rare species are falling and then counts the total number of rare species in each grid cell.

To quantify how the observed number of rare species differ from the null expectation we then use Standardized Effect Sizes (SES) as follows:

$$SES = (X - Y)/Z$$

X as the number of rare species observed in each grid cell,

Y as the average of rare species found from the null model after 2000 interactions and

Z as the standard deviation from Y.

A positive SES indicates more rare species than would be expected by chance and a negative SES fewer than expected.

Since we are using 14 different systems (7 Coastal Systems and 7 High Seas (see Extended Data Fig. 3)) for 2 groups of organisms (bony and cartilaginous fish), 2 functional rarity indices (distinctiveness and uniqueness), and using 3 different probabilities of occurrences (prob. > 0.9, > 0.7 and > 0.5).

We then have the following “roadmap”:

- i. Scales – 7 Coastal Systems and 7 High Seas.
- ii. Groups – bony and cartilaginous fish.
- iii. Indices – distinctiveness and uniqueness.
- iv. Probability of occurrences – > 0.9, > 0.7 and > 0.5.
- v. Total – 168 independent cases analysis (each having its own assemblage and trait matrices).

Therefore, as mentioned above, the null model ran 2000 interactions for each of those independent cases. The final matrices from these initial steps contain grid cells as rows and as columns we have the raw number of rare species along with the SES values for each. These matrices were important to map our results.”

The Figure 1 a and c respectively shows the biogeographic distribution of raw rarity for bony fish and cartilaginous in Coastal Systems. The grid cells with concentrations of rare species higher than expected by chance are in red and lower than expected in blue (Figure 1 b and d). These results are also in concordance with our previous findings.

(see also Figures 2 (for High Seas), and Extended Data Fig. 4 and 5 (for complementary functional index - uniqueness).

Line 79: “we compared these results (Extended Data Fig. 4) with an alternative method of calculating rarity (using functional uniqueness (Extended Data Fig. 2) rather than functional distinctiveness, see Extended Data Fig. 5 to 7) to ensure the robustness of our approach; the conclusions were consistent (see Extended Data Fig. 6 and 7).”

I could not find support for these statements in the results. First, Extended Data Fig. 2 shows a correlation very close to zero between distinctiveness and uniqueness for the two groups of fishes. This means that distinctiveness and uniqueness are complementary sources of information about functional rarity. Second, we can see much more areas with rarity in Extended Data Fig. 6 than in Fig. 1. Therefore, results were sensitive to the functional index being used.

Answer:

Thank you for raising this point. We present the results based on distinctiveness in the Figure 1 and 2 and the results based on uniqueness in the Extended Data Fig. 4 and 5.

Line 83: “It is possible that a sampling bias towards records in coastal waters could partly explain our findings. However, we note that regions of the earth that are typically under-recorded in biodiversity surveys, such as South America and Africa, are identified in our analysis as important repositories of rarity, while marine areas that have been intensively monitored, such as the coasts of NW Europe are not.”

This might be an artifact of using rarity ~ richness residuals, as mentioned above. I would consider the use of null models.

Answer:

As we explained above, now we have applied a null model approach and are using the SES to investigate how the number of rare species in a given grid cell differ from the null expectation. We are pleased to report that our previous findings were supported using this approach.

Line 106: Why discussing patterns of biodiversity in land?

Answer:

To provide context, in relation to conservation decisions. We believe that terrestrial and aquatic biodiversity share similar challenges regarding coverage by protected areas.

Fig. 1: it is hard to see the contrast between cells, as well fish silhouettes (mainly in ‘d’). Why not use black silhouettes?

Answer:

Done.

Extended Data Fig. 1 – not enough resolution to see numbers and formulae.

Answer:

We have endeavoured to improve this.

Line 192: “Extended Data Fig. 3 c and d, these traits are largely uncorrelated.”

I think this figure means exactly the opposite. If the relation is maintained if you remove 1 trait, then the information removed is still in the remaining traits.

Answer:

The Extended Data Fig. 8 from a to d show the sensitivity analysis results; the strong correlations reported are a good indication that the environmental traits are not driving our results. Meanwhile Extended Data Fig. 8 e and f show the result from the trait correlation analysis (with very weak correlations around zero, which is a good indication that all traits are largely uncorrelated). We hope that now it is better explained.

Line 542: “Functional distinctiveness is measured by the average of the functional distance of a given species compared with all other species in the regional pool”.

Does it mean that, for example, a species from Australia was compared to a species from UK? I would impose more restrictions to this “regional pool” (which is truly the global pool of fish species). Perhaps you should restrict Functional distinctiveness calculations within the pool of biogeographic (e.g., oceanic) regions (like you showed in Extended Data Table 1).

Answer:

We thank the reviewer for this comment. As suggested, we performed our analyses for each of the seven Oceanic Regions individually (those are: North Atlantic, South Atlantic, Indian Ocean, North West Pacific, South West Pacific, North East Pacific and South East Pacific (Extended Data Fig 3). We ran these analyses separately for the Coastal Systems and High Seas from those seven Oceanic Regions. As in the previous analyses, the species were split by bony fishes and cartilaginous fishes. These new results support our previous conclusion that there are higher concentrations of rare species towards higher latitudes and near the continents.

Line 559: Extended Data Fig. 3. It is lacking Y-axis labels. Would it be frequency?

Answer:

We have changed and updated all figures. Thank you.

Line 562: Extended Data Fig. 2. Distribution of correlations between trait values for the Actinopterygii (c) and Elasmobranchii (d). Correlations are weak and largely centered on zero, particularly for the Actinopterygii, indicating that traits provide complementary information on function”

It means that uniqueness and distinctiveness provide complementary information about rarity. As there is no correlation at all there’s no reason to show the results for only one functional index.

Answer:

We now show the results found for functional uniqueness in the Extended Data Fig. 4 and 5.

Line 567: Extended Data Fig. 4 – The distribution of restrictedness is very skewed, with a large number of species presenting high restrictedness; that’s why your horizontal red line is very close to 1. That’s because you have a very large total number of cells relative to number of cells occupied by rare species in the cells. I would consider other ways to calculate GE (see comments in the beginning of this letter).

Answer:

As explained above, we have now split the oceans into Coastal Systems (Figure 1) and High Seas (Figure 2). In addition, we used a smaller grid cell size and ran all the analysis using different probabilities of occurrences (Extended Data Fig. 3). We hope that all those steps will persuade the reviewer about the robustness of our approach.

Line 581: Extended Data Fig. 6 – one silhouette is superimposing “residuals” in d

Answer:

All corrected. Thanks.

Supplementary table 2: please rank your models by increasing AIC values (or instead by AICw, which you must show as it depicts the weight of one model relative to the other models).

Answer:

Thank you. We no longer include the models in this current paper as the additional analyses we now report provide a more coherent story. We felt that this modelling would be better placed in another paper where there would be sufficient opportunity to delve into it with the detailed attention it deserves.

References used in this letter:

- 1 Strona, G., Nappo, D., Boccacci, F., Fattorini, S. & San-Miguel-Ayanz, J. A fast and unbiased procedure to randomize ecological binary matrices with fixed row and column totals. *Nature communications* **5**, 1-9 (2014).
- 2 Kaschner, K. *et al.* AquaMaps: Predicted range maps for aquatic species. World wide web electronic publication, www.aquamaps.org, Version 10/2019., 2019).
- 3 Froese, R. & Pauly, D. Editors. FishBase. World Wide Web electronic publication, www.fishbase.org, version (02/2021). <www.fishbase.org, (02/2021)> (2021).
- 4 Signorell, A. *et al.* DescTools: Tools for descriptive statistics. *R package version 0.99* **18** (2016).
- 5 Rabinowitz, D. (John Wiley and Sons: Chichester, UK, 1981).
- 6 Grenié, M., Denelle, P., Tucker, C. M., Munoz, F. & Violle, C. funrar: An R package to characterize functional rarity. *Diversity and Distributions* **23**, 1365-1371, doi:10.1111/ddi.12629 (2017).
- 7 Gaston, K. J. in *Rarity. Population and Community Biology Series* Vol. 13 1-21 (Springer, 1994).

Reviewer comments, second round review:

Reviewer 1

Thank you very much for giving me the opportunity to review a revised version of the paper "Biogeography of rare fish in the world's oceans". I appreciate the Authors' efforts and the additional analyses they performed to deal with the reviewers' comments. The main highlights of their review is the adoption of a higher spatial resolution, the division of the results in different biogeographical regions, the replacement of the regression residual approach with one based on null model analysis, and the performance of sensitivity analyses. I think the analyses are informative and add to the paper, but I have still some concerns about the methodological approach.

The use of different biogeographical regions and the sensitivity analyses were performed by the Authors in response (amongst the others) to the main concern I expressed about their analyses during the first round of reviews, which I am pasting below. However, from my understanding of this new version, I am not 100% sure that they have actually tackled the issue. In fact, my main point was about the potential circularity introduced by computing rarity of each species using the full set of fish species. It seems to me – but I might be wrong – that the Authors kept the computation of species' rarity as in the first version (pooling all species), and used the biogeoregions only thereafter (this seems evident, for example, from Extended Data Fig. 8, where uniqueness and distinctiveness are presented for all species worldwide, and not by region). Again, the MS was a bit unclear about this point, so apologies if I am wrong. However, if my interpretation is correct, the Authors' approach does not resolve the doubts raised by my comments; my suggestion was to look at patterns of rarity by subsetting fish species by (for example) bioclimatic regions before computing their rarity. The sensitivity analyses are somehow helpful, but my remark was about the rarity of some combination of species' features representing some rare combinations of environmental/ecological conditions not necessarily captured in the variables used in the analyses (see the pasted text below for the full reasoning). It is a subtle point but, in my opinion, a crucial one. Incidentally, I understand and appreciate the double-blind peer review process, but I would have liked to have a look at the code during the review (as this would have, for instance, clarified these doubts). This could have been provided in anonymous form as the rest of the paper.

My comments from the first round of reviews:

"I am not 100% I understood this correctly, but my impression is that the Authors assessed rarity by looking at the complete set of fish species (separately for bony and cartilaginous fish). I suspect that this, combined with the choice of the environmental traits (especially maximum depth, mean temperature preference) and some adaptive traits (such as swimming mode, body shape), might create some circularity. That is, the measure of global rarity for a given species might partly reflect the availability of habitat/conditions suitable to the species. Then, I might expect to have more niches for rare species where the local environmental setting is more rare.

To make an extreme example, I can imagine that a specific set of environmental/ecological conditions can be found only in a few localities worldwide. Some species might be well adapted to those particular conditions. Therefore, their range would be likely restricted to that locality, and their functional traits would also reflect the peculiarity of the environment. The Authors' approach would therefore identify those species as rare and the area where the species occur as a hotspot of rarity. This is fine, but I think that it would be quite important to discriminate between this kind of situations (where local rarity is a consequence of local environmental/ecological features) and a situation where rarity depends only on species. From a conservation perspective this is very important (as the first case would call for management strategies focusing on the site; while the latter would require strategies focusing on the species).

The confounding effect of the role of the environment on species traits (and hence the potential link between global patterns of environmental variability and global patterns of rarity) could be possibly reflected in the large scale overall results. Especially, the main findings of more rarity in the coastal areas and at higher latitudes seem to me as potential consequences of this mechanism. Coastal environments clearly provide more environmental heterogeneity (and hence possibly more specialized habitats promoting the presence of rare species according to the Authors' metric) compared to the pelagic realm. The pattern of increasing rarity with latitude might be even more an effect of the analytical approach: by pooling all species together when calculating rarity, given

the higher fish diversity of the tropics and the selectivity of the high-latitude environments, it is not surprising to find more rare species in temperate and cold-waters realms than in the tropics (which in turn drives the observed patterns). The Authors tried to control for this effect, but see my comments below on their approach.

One potential solution to the issue is that of examining patterns of rarity by splitting the global species pool into subsets based on some biogeographical/environmental criteria. The issue was considered also in the aforementioned paper about reef fish functional rarity ("We tested the influence of the species pool definition from which functional distinctiveness was extracted. To this aim, we split the cells into two realms—Atlantic and Pacific—and estimated species distinctiveness values separately in each realm").

Selecting a different set of functional traits not reflecting adaptation to specific habitat conditions could be another potential strategy, but I am not sure which traits I would use. Perhaps removing at least the mean temperature preference and/or the max depth could be one starting point, but I guess the issue is a bit philosophical as every single trait could be seen as an adaptation to some environment and I cannot see any strong rule or argument to identify traits that are more or less important to define a species rarity."

It seems to me that a similar remark was also made by the other Reviewer, when she/he writes: "Functional distinctiveness/uniqueness were calculated based on a global rather than a regional pool of species. In the present calculation, one fish from e.g., Australia can be compared to a fish from e.g., UK. Fishes from different regions likely will have different traits because they have quite different evolutionary histories. I would include in my regional pool the species found in each Oceanic region"

The Authors write that they did so, but again my feeling is that they computed rarity based on the global pool, and only then projected results in the different regions. Sorry if I misunderstood this important point!

Both Reviewer 2 and I highlighted the existence of different facets of rarity (see Rev.2's comment about Rabinowitz's concept of rarity). In particular, I wrote:

"To this end, it would also be important to consider that the concept of rarity is a very relative one (while the Authors seem to present it as a quite absolute measure). To this regard, I think Table 1 in the perspective paper mentioned above (<https://doi.org/10.1016/j.tree.2017.02.002>) provides a valuable roadmap to the different possible types (12) of functional rarity. I am not suggesting that the Authors need to explore all the possible types of rarity, but at least they should clarify where their measure falls into the classification scheme."

The Authors did not reply to this comment, and I think they also did not tackle Rev.2's suggestion (which was, in my interpretation, about computing additional measures of rarity referring to abundance and habitat specialization, and then compare them; instead, the Authors stuck to their original measure of rarity and then: (1) explored how those varied for 3 different thresholds of probability occupancy; and (2) compared the distribution of habitat specialization distribution in rare vs. non rare species. These analyses are informative, but not revealing the different rarity facets. Perhaps the available data (or better, lack of) makes this difficult, but the topic needs to be addressed/discussed. The suggested use of probability values as proxies for abundance appears as, perhaps, the only viable option considering the global scale of the analysis. However, I have some concerns in using the threshold alone as a proxy for abundance. There might be sound ways to use it in combination with species life traits (e.g. body size, trophic level, growth rate? For the same probability of occurrence, I would expect much less individuals of a large predatory species compared to a small planktivore), but perhaps I would not incorporate these in the analyses, while I would state more clearly all the potential caveats associated to this topic.

Regarding the use of probability thresholds, Reviewer 2 made a good suggestion when she/he wrote:

"Hence authors should consider using maps with probability thresholds lower than 0.9. To avoid subjective decisions about probability thresholds, I would calculate GE as $GE = 1 - (\text{sum of probabilities of species } i / \text{total number of cells in region } r)$."

This seems an elegant solution to me, but the Authors decided to use 3 different thresholds instead; the consistency between the results obtained with the different thresholds is of course good news, but the approach seems suboptimal to me compared to the Reviewer's suggestion (and has the drawbacks of triplicating the output).

Finally, I think the null model is well implemented and a much better approach than the previous one based on the regression residuals.

Reviewer #4 (Remarks to the Author):

Manuscript - Biogeography of rare fish in the world's oceans

This manuscript combined different datasets and trait species to investigate the rarity patterns of fish and their overlap with marine protected areas. This study is of broad interest to biologists and ecologists since the authors explored several measures of rarity to favor an integrative knowledge on the processes that structure fish fauna (bony and cartilaginous fish). On the other hand, after reviewing all comments and suggestions performed in the previous review round, I observed that all comments were answered satisfactorily. However, I have some new comments that maybe can help improve more the quality of the manuscript. For instance, the authors mentioned that two traits were removed (i.e., maximum depth and preference temperature) to test the sensitivity in the analysis. Although this approach is correct, I have a concern about if the authors will be found similar patterns removing some categorical traits, which have different levels and weights within the analysis. Thus, I suggest testing the sensibility of removing categorical traits. Finally, the authors mention in line 120 that there is a sampling bias in regions, such as South America and Africa, this sentence is not true considering studies such as Floeter et al. 2007 and Pinheiro et al. 2018, as well as online repositories, such as Reef Life Survey, Fishes Greater Caribbean and Fishes East Pacific. Thus, my suggestion is to re-write this sentence.

ANSWERS TO THE REVIEWERS COMMENTS

Reviewer #1 (Remarks to the Author):

Dear Editor,

Thank you very much for giving me the opportunity to review a revised version of the paper "Biogeography of rare fish in the world's oceans". I appreciate the Authors' efforts and the additional analyses they performed to deal with the reviewers' comments. The main highlights of their review is the adoption of a higher spatial resolution, the division of the results in different biogeographical regions, the replacement of the regression residual approach with one based on null model analysis, and the performance of sensitivity analyses. I think the analyses are informative and add to the paper, but I have still some concerns about the methodological approach.

The use of different biogeographical regions and the sensitivity analyses were performed by the Authors in response (amongst the others) to the main concern I expressed about their analyses during the first round of reviews, which I am pasting below. However, from my understanding of this new version, I am not 100% sure that they have actually tackled the issue. In fact, my main point was about the potential circularity introduced by computing rarity of each species using the full set of fish species. It seems to me – but I might be wrong - that the Authors kept the computation of species' rarity as in the first version (pooling all species), and used the biogeoregions only thereafter (this seems evident, for example, from Extended Data Fig. 8, where uniqueness and distinctiveness are presented for all species worldwide, and not by region). Again, the MS was a bit unclear about this point, so apologies if I am wrong. However, if my interpretation is correct, the Authors'

approach does not resolve the doubts raised by my comments; my suggestion was to look at patterns of rarity by subsetting fish species by (for example) bioclimatic regions before computing their rarity. The sensitivity analyses are somehow helpful, but my remark was about the rarity of some combination of species' features representing some rare combinations of environmental/ecological conditions not necessarily captured in the variables used in the analyses (see the pasted text below for the full reasoning). It is a subtle point but, in my opinion, a crucial one. Incidentally, I understand and appreciate the double-blind peer review process, but I would have liked to have a look at the code during the review (as this would have, for instance, clarified these doubts). This could have been provided in anonymous form as the rest of the paper.

Answer: We would like to thank the reviewer for raising this important issue. There are two important points here, namely the information contained in ED fig 8, and the analyses presented in the body of the paper.

1. We apologise for the fact that the legend from the Extended Data Fig. 8 in the previous version of the ms was lacking some important information: this figure represents the sensitivity analysis performed only for the Northwest Pacific Ocean coastal system. To avoid further confusion, we decided to split the previous Extended Data Fig. 8 (a to f) into two separated figure. **Extended Data Fig. 8** (a to d): showing only the sensitivity analysis (using the Northwest Pacific Ocean as an example), and **Extended Data Fig. 9** (a and b): showing only the trait correlation analysis (this was done a priori, to ensure the complementarity of all 11 traits used).

2. We stress that all the analyses in the main paper were done for each “independent case” separately. Between lines 293 and 303, we provide a “roadmap” explaining each of these “independent cases”, as follows:

“Since we are using 14 different systems (7 Coastal Systems and 7 High Seas (see Extended Data Fig. 3)) for 2 groups of organisms (bony and cartilaginous fish), 2 functional rarity indices (distinctiveness and uniqueness), and using 3 different probabilities of occurrences (prob. > 0.9, > 0.7 and > 0.5).

We then have the following “roadmap”:

- i. Scales – 7 Coastal Systems and 7 High Seas.*
- ii. Groups – bony and cartilaginous fish.*
- iii. Indices – distinctiveness and uniqueness.*
- iv. Probability of occurrences – > 0.9, > 0.7 and > 0.5.*
- v. Total – 168 **independent cases** analysis (each having its own assemblage and trait matrices.”*

We further wish to emphasise that the null model ran 2000 iterations for each of those 168 independent cases.

In additional response to the reviewer comment, we are not pooling all species together to compute the rarity indices; instead, we take the 168 independent cases and compute the metrics separately for each of them. For example, we compute distinctiveness, uniqueness and restrictedness for the species from the North Atlantic coastal system, then repeat the process for the species from the North Atlantic high seas (open ocean), and then for each of the other oceanic regions (one system at a time, keeping coastal systems and high seas separate in all cases).

My comments from the first round of reviews:

“I am not 100% I understood this correctly, but my impression is that the Authors assessed rarity by looking at the complete set of fish species (separately for bony and cartilaginous fish). I suspect that this, combined with the choice of the environmental traits (especially maximum depth, mean temperature preference) and some adaptive traits (such as swimming mode, body shape), might create some circularity. That is, the measure of global rarity for a given species might partly reflect the availability of habitat/conditions suitable to the species. Then, I might expect to have more niches for rare species where the local environmental setting is more rare. To make an extreme example, I can imagine that a specific set of environmental/ecological conditions can be found only in a few localities worldwide. Some species might be well adapted to those particular conditions. Therefore, their range would be likely restricted to that locality, and their functional traits would also reflect the peculiarity of the environment. The Authors' approach would therefore identify those species as rare and the area where the species occur as a hotspot of rarity. This is fine, but I think that it would be quite important to discriminate between this kind of situations (where local rarity is a consequence of local environmental/ecological features) and a situation where rarity depends only on species. From a conservation perspective this is very important (as the first case would call for management strategies focusing on the site; while the latter would require strategies focusing on the species).

The confounding effect of the role of the environment on species traits (and hence the potential link between global patterns of environmental variability and global patterns of rarity) could be possibly reflected in the large scale overall results. Especially, the main findings of more rarity in the coastal areas and at higher latitudes seem to me as potential consequences of this mechanism. Coastal environments clearly provide more environmental heterogeneity (and hence possibly

more specialized habitats promoting the presence of rare species according to the Authors' metric) compared to the pelagic realm. The pattern of increasing rarity with latitude might be even more an effect of the analytical approach: by pooling all species together when calculating rarity, given the higher fish diversity of the tropics and the selectivity of the high-latitude environments, it is not surprising to find more rare species in temperate and cold-waters realms than in the tropics (which in turn drives the observed patterns). The Authors tried to control for this effect, but see my comments below on their approach.

One potential solution to the issue is that of examining patterns of rarity by splitting the global species pool into subsets based on some biogeographical/environmental criteria. The issue was considered also in the aforementioned paper about reef fish functional rarity (“We tested the influence of the species pool definition from which functional distinctiveness was extracted. To this aim, we split the cells into two realms—Atlantic and Pacific—and estimated species distinctiveness values separately in each realm”).

Selecting a different set of functional traits not reflecting adaptation to specific habitat conditions could be another potential strategy, but I am not sure which traits I would use. Perhaps removing at least the mean temperature preference and/or the max depth could be one starting point, but I guess the issue is a bit philosophical as every single trait could be seen as an adaptation to some environment and I cannot see any strong rule or argument to identify traits that are more or less important to define a species rarity.”

It seems to me that a similar remark was also made by the other Reviewer, when she/he writes: “Functional distinctiveness/uniqueness were calculated based on a global rather than a regional pool of species. In the present calculation, one fish from e.g., Australia can be compared to a fish from e.g., UK. Fishes from different regions likely will have different traits because they have quite different evolutionary histories. I would include in my regional pool the species found in each Oceanic region”

The Authors write that they did so, but again my feeling is that they computed rarity based on the global pool, and only then projected results in the different regions. Sorry if I misunderstood this important point!

Answer: Thank you for commenting on this important point. We are now provide the code and data to show how we computed the indices. These computations were performed at a local scale, and, as noted above, we analysed each of the 168 independent cases separately.

Both Reviewer 2 and I highlighted the existence of different facets of rarity (see Rev.2's comment about Rabinowitz's concept of rarity). In particular, I wrote:

“To this end, it would also be important to consider that the concept of rarity is a very relative one (while the Authors seem to present it as a quite absolute measure). To this regard, I think Table 1 in the perspective paper mentioned above ([MailScanner has detected a possible fraud attempt from "urldefense.proofpoint.com" claiming to be https://doi.org/10.1016/j.tree.2017.02.002](https://doi.org/10.1016/j.tree.2017.02.002)) provides a valuable roadmap to the different possible types (12) of functional rarity. I am not suggesting that the Authors need to explore all the possible types of rarity, but at least they should clarify where their measure falls into the classification scheme.”

Answer: Thank you for this suggestion. We have now included a new section in our methods, as follows (lines between 367 and 376):

“Forms of Functional Rarity Classification Scheme

In their 2017 paper Violle *et al.*, suggested 12 forms of functional rarity. We believe that the approach applied here is most similar to the classification scheme they described as : “Rare traits irrespective of the scale and the species pool”. The authors pointed to two possible extremes: rare traits (exhibited by range-restricted species) and common traits (supported by many widespread species). In this case, our approach identifies species that are both geographically restricted within each of the 14 systems (coastal and high seas systems) and present a distinct (or unique) combination of traits. Our approach to the classification of rarity differs slightly, however, in that we follow Gaston’s approach of quantile distribution as illustrated in Extended Data Fig. 1, step 3 QUARTILES.”

The Authors did not reply to this comment, and I think they also did not tackle Rev.2’s suggestion (which was, in my interpretation, about computing additional measures of rarity referring to abundance and habitat specialization, and then compare them; instead, the Authors stuck to their original measure of rarity and then: (1) explored how those varied for 3 different thresholds of probability occupancy; and (2) compared the distribution of habitat specialization distribution in rare vs. non rare species. These analyses are informative, but not revealing the different rarity facets. Perhaps the available data (or better, lack of) makes this difficult, but the topic needs to be addressed/discussed. The suggested use of probability values as proxies for abundance appears as, perhaps, the only viable option considering the global scale of the analysis. However, I have some concerns in using the threshold alone as a proxy for abundance. There might be sound ways to use it in combination with species life traits (e.g. body size, trophic level, growth rate? For the same probability of occurrence, I would expect much less individuals of a large predatory species compared to a small planktivore), but perhaps I would not incorporate these in the analyses, while I would state more clearly all the potential caveats associated to this topic.

Answer: We would like to thank the reviewers for the opportunity of addressing this important topic. We have now added the following paragraph at the end of the main text (lines between 125 and 139).

“AquaMaps provides us with a unique opportunity to use occurrence data for measuring rarity based on occupancy data. We treated species occurrences in each of the two-degree grid cells as a proxy for rarity and used this information to distinguish between rare species (highly restricted species) and common species (widely occurring species). The global coverage of AquaMaps is an advantage here. There are, however, some caveats associated with our approach. For example, the data provided by AquaMaps include not only actual observations but also probabilities of occupancy based on combinations of sightings and internal algorithms. Moreover, AquaMaps is based on summed information rather than being a representation of the species present at a given point in time. Additionally, the nature of our data means we cannot calculate shifts in rarity on a temporal scale or with any weighting by abundances. We also appreciate that when abundance and/or biomass data are available, another taxonomic rarity metric, namely Taxonomic Scarcity, could be used. Our decision to focus on restrictedness was shaped by the fact that we are using geographical information from all described fish available in AquaMaps, and that this database provides occupancy patterns based on different probabilities of occurrences.”

Regarding the use of probability thresholds, Reviewer 2 made a good suggestion when she/he wrote:

“Hence authors should consider using maps with probability thresholds lower than 0.9. To avoid subjective decisions about probability thresholds, I would calculate GE as $GE = 1 - (\text{sum of probabilities of species } i / \text{total number of cells in region } r)$.”

This seems an elegant solution to me, but the Authors decided to use 3 different thresholds instead; the consistency between the results obtained with the different thresholds is of course good news, but the approach seems suboptimal to me compared to the Reviewer’s suggestion (and has the drawbacks of triplicating the output).

Answer: We thank the reviewer for this suggestion. We were uncertain about how to implement this alternative approach to computing GE and so consulted a statistician colleague for advice. He also had questions about its implementation, and felt that it might be less informative than our initial approach (albeit with the benefit of producing fewer plots in the supplementary material). We therefore decided against adopting the suggested solution and instead have retained our initial approach which examines alternative probabilities. We highlight the fact that this approach yielded consistent results with our initial analyses and that it is accessible to readers of the paper.

Finally, I think the null model is well implemented and a much better approach than the previous one based on the regression residuals.

Reviewer #4 (Remarks to the Author):

Manuscript - Biogeography of rare fish in the world’s oceans

This manuscript combined different datasets and trait species to investigate the rarity patterns of fish and their overlap with marine protected areas. This study is of broad interest to biologists and ecologists since the authors explored several measures of rarity to favor an integrative knowledge on the processes that structure fish fauna (bony and cartilaginous fish). On the other hand, after reviewing all comments and suggestions performed in the previous review round, I observed that all comments were answered satisfactorily. However, I have some new comments that may help improve the quality of the manuscript. For instance, the authors mentioned that two traits were removed (i.e., maximum depth and preference temperature) to test the sensitivity in the analysis. Although this approach is correct, I have a concern about if the authors will be found similar patterns removing some categorical traits, which have different levels and weights within the analysis. Thus, I suggest testing the sensitivity of removing categorical traits.

Answer: The results from this new sensitivity analysis are now included as Extended Data Fig. 8. In this figure, we present the results found for the Coastal System of the Northwest Pacific Ocean as an example. We first computed distinctiveness and uniqueness using all eleven traits. Then, we recomputed the indices removing the categorical traits (and environmental traits) one by one. Next, we calculated the Spearman correlations between the indices found using all traits by indices found using all traits minus one trait. We now also include a table showing the results of those correlations; this underlines the robustness of our analysis. As you can see, all correlations are above 0.75413. As such, we are confident that the results found are not just a reflection of one of the traits driving the results.

Finally, the authors mention in line 120 that there is a sampling bias in regions, such as South America and Africa, this sentence is not true considering studies such as Floeter et al. 2007 and Pinheiro et al. 2018, as well as online repositories, such as Reef life Survey, Fishes Greater Caribbean and Fishes East Pacific. Thus, my suggestion is to re-write this sentence.

Answer: Thank you. We have now included those citations (lines 114 and 115) as exceptions that can be made regarding fish biodiversity sampling collections at the Neotropical area. Our objective with this statement was just to say that even though areas such as the NW coasts of Europe are the best sampled areas of the earth, our results identify rarity concentrations in areas that are less well sampled.

Reviewer comments, third round review:

Reviewer 1

Reviewer #1 (Remarks to the Author):

I have read the revised version of the MS and I am now satisfied with the revision. I have a couple of final suggestions which might further increase the clarity of the MS and the robustness of the results.

Now it is clear from the code and response to reviewers that the Authors have indeed computed rarity separately for each subregion, i.e. also subsetting the functional traits. However, I would add a sentence at around line 57 to make this explicit. After the sentence "Moreover, our investigation was conducted separately for each Coastal System and High Seas in seven Oceanic Regions (14 different systems in total)." I would add something like: "That is, for each region, we assessed species rarity based on the distribution and functional traits of species present in that region..." etc.

It is also an interesting point for discussion that, with this approach (which makes sense to me), a species might have a widespread distribution and globally redundant traits but be locally rare in a specific region. Conversely, while endemic species will be of course always rare. This generates different scenarios for conservation/diversity loss studies. If a widespread species locally identified as rare goes locally extinct, there will be consequences in the overall functional space, but those might be neutralized by recolonization from the other regions where the species exists. Conversely, an endemic species could not be rescued. It would be worth discussing this aspect, ideally providing some number (even a map?) to quantify how many species identified as rare are/are not endemic (and where?).

One final suggestion regards a potentially important point regarding the effect of sampling biases on the estimation of rarity; I am sorry I spotted this only at this late stage, but it was a non-obvious issue requiring a little bit of thinking. In lines 112-117, the Authors seem to imply that undersampled areas should be expected to show low rarity compared to rich ones, while they found rarity in poorly investigated areas but not in highly investigated areas, as a circumstantial support to their analyses not being biased by sampling effort. However, I think the issue is more subtle, and related also to the null model analysis. I wrote some toy code to check whether the fill of a random species x locality matrix affects the estimation of rare species (based only on their distribution) when the fixed-fixed null model selected by the Authors is applied. From my simplified experiments (I am pasting the code below), it seems to me that matrix fill (which is obviously dependent on study effort, but might also be a plain consequence of differences in regional patterns of species richness) does not affect the null-model-based rarity estimation; however, that might be a consequence of the random nature of the toy matrices. It would be interesting if you could plot the number of hotspots per regional (fish sp x cell) matrix vs. matrix fill, as that might be more indicative of potential biases.

In making these experiments, I also asked myself whether the identification of rare species was redone each time the null model was applied, but then I realized that this does not matter, as the fixed-fixed null model preserve the number of locality where each species occur, so the species range facet of the rarity measure is not affected. Perhaps it might be worth making this point explicit, just in case some readers interested in technicalities ask themselves the same question.

```
#####R Code
```

```
###curveball algorithm for null model
FF<-functon(m){
RC=dim(m)
R=RC[1]
C=RC[2]
```

```

hp=list()
for (row in 1:dim(m)[1]) {hp[[row]]=(which(m[row,]==1))}
l_hp=length(hp)
for (rep in 1:(5*l_hp)){
AB=sample(1:l_hp,2)
a=hp[[AB[1]]]
b=hp[[AB[2]]]
ab=intersect(a,b)
l_ab=length(ab)
l_a=length(a)
l_b=length(b)
if ((l_ab %in% c(l_a,l_b))==F){
tot=setdiff(c(a,b),ab)
l_tot=length(tot)
tot=sample(tot, l_tot, replace = FALSE, prob = NULL)
L=l_a+l_ab
hp[[AB[1]]] = c(ab,tot[1:L])
hp[[AB[2]]] = c(ab,tot[(L+1):l_tot])}
}
rm=matrix(0,R,C)
for (row in 1:R){rm[row,hp[[row]]]=1}
rm
}

```

####Example

```

mat<-matrix(runif(100),10)
fill<-0.4
mat1<-1*(mat>fill)

```

###species are rows, columns are localities
###compute species rarity based on percentile

###number of localities where each species is found
loc_n<-rowSums(mat1)

###identify threshold for rarity
rar_tre<-quantile(loc_n,0.75)

###non rare species
nr_spp<-which(loc_n<rar_tre)

###number rare species per loc
rar_mat<-mat1
rar_mat[nr_spp,]<-0
r_spp_n<-colSums(rar_mat)

###null model
nm_sc<-c()
for (rep in 1:100){
nm_rar_mat<-FF(mat1)
nm_rar_mat[nr_spp,]<-0
nm_sc<-rbind(nm_sc,colSums(nm_rar_mat))
}

```

nm_mean<-apply(nm_sc,2,'mean')
nm_sd<-apply(nm_sc,2,'sd')
z<-(r_spp_n-nm_mean)/nm_sd
#number of significant rare species
sum(z>2)

###now explore the effect of matrix fill (~ sampling intensity)

mat<-matrix(runif(300),30)

fill_vals<-seq(0.2,0.9,0.01)
hs<-c()
for (fill in fill_vals){
mat1<-1*(mat>fill)
loc_n<-rowSums(mat1)
rar_tre<-quantile(loc_n,0.75)
nr_spp<-which(loc_n<rar_tre)
rar_mat<-mat1
rar_mat[nr_spp,]<-0
r_spp_n<-colSums(rar_mat)
nm_sc<-c()
for (rep in 1:100){
nm_rar_mat<-FF(mat1)
nm_rar_mat[nr_spp,]<-0
nm_sc<-rbind(nm_sc,colSums(nm_rar_mat))
}
nm_mean<-apply(nm_sc,2,'mean')
nm_sd<-apply(nm_sc,2,'sd')
z<-(r_spp_n-nm_mean)/nm_sd
#number of significant rare species
hs<-c(hs,mean(z,na.rm=T)) #explore average z; can be replaced with sum(z>2) for example
}

plot(fill_vals,hs)

```

REVIEWERS' COMMENTS

Reviewer #1 (Remarks to the Author):

I have read the revised version of the MS and I am now satisfied with the revision. I have a couple of final suggestions which might further increase the clarity of the MS and the robustness of the results.

Now it is clear from the code and response to reviewers that the Authors have indeed computed rarity separately for each subregion, i.e. also subsetting the functional traits. However, I would add a sentence at around line 57 to make this explicit. After the sentence "Moreover, our investigation was conducted separately for each Coastal System and High Seas in seven Oceanic Regions (14 different systems in total)." I would add something like: "That is, for each region, we assessed species rarity based on the distribution and functional traits of species present in that region..." etc.

Answer: We thank the reviewer for this suggestion. We now have included the above sentence after that statement. See lines from 69 to 72.

It is also an interesting point for discussion that, with this approach (which makes sense to me), a species might have a widespread distribution and globally redundant traits but be locally rare in a specific region. Conversely, while endemic species will be of course always rare. This generates different scenarios for conservation/diversity loss studies. If a widespread species locally identified as rare goes locally extinct, there will be consequences in the overall functional space, but those might be neutralized by recolonization from the other regions where the species exists. Conversely, an endemic species could not be rescued. It would be worth discussing this aspect, ideally providing some number (even a map?) to quantify how many species identified as rare are/are not endemic (and where?).

Answer: Thank you for this suggestion, we are now including a list of rare species found for all fourteen independent systems, for bony and cartilaginous fish. See Supplementary Table 2 in the Supplementary Information file.

One final suggestion regards a potentially important point regarding the effect of sampling biases on the estimation of rarity; I am sorry I spotted this only at this late stage, but it was a non-obvious issue requiring a little bit of thinking. In lines 112-117, the Authors seem to imply that undersampled areas should be expected to show low rarity compared to rich ones, while they found rarity in poorly investigated areas but not in highly investigated areas, as a circumstantial support to their analyses not being biased by sampling effort. However, I think the issue is more subtle, and related also to the null model analysis. I wrote some toy code to check whether the fill of a random species x locality matrix affects the estimation of rare species (based only on their distribution) when the fixed-fixed null model selected by the Authors is applied.

From my simplified experiments (I am pasting the code below), it seems to me that matrix fill (which is obviously dependent on study effort, but might also be a plain consequence of differences in regional patterns of species richness) does not affect the null-model-based rarity estimation; however, that might be a consequence of the random nature of the toy matrices. It would be interesting if you could plot the number of hotspots per regional (fish sp x cell) matrix vs. matrix fill, as that might be more indicative of potential biases.

In making these experiments, I also asked myself whether the identification of rare species was redone each time the null model was applied, but then I realized that this does not matter, as the fixed-fixed null model preserve the number of locality where each species occur, so the species range facet of the rarity measure is not affected. Perhaps it might be worth making this point explicit, just in case some readers interested in technicalities ask themselves the same question.

Answer: Thank you for all your suggestions towards the improvements of our paper. The identification of rare species was not redone each time when the null model was applied. Based on the reviewers' suggestions and code we tested whether or not the fill of a random species by location cell matrix affects the estimation of rare species by distribution when the null model is applied, and we found no bias. A new section was included in the Supplementary Information file to demonstrate the results we found using four different matrices: North West Pacific Coast (bony and cartilaginous species) and South West Pacific Coast (bony and cartilaginous species). See the Supplementary Information file: Supplementary Figure 11, under Supplementary Analysis section. A new section was also included on the MS methods, between lines 366 and 371, with a comment at the main text (see lines 133 and 134).

#####R Code

```
###curveball algorithm for null model
FF<-functon(m){
RC=dim(m)
R=RC[1]
C=RC[2]
hp=list()
for (row in 1:dim(m)[1]) {hp[[row]]=(which(m[row,]==1))}
l_hp=length(hp)
for (rep in 1:(5*l_hp)){
AB=sample(1:l_hp,2)
a=hp[[AB[1]]]
b=hp[[AB[2]]]
ab=intersect(a,b)
l_ab=length(ab)
l_a=length(a)
l_b=length(b)
if ((l_ab %in% c(l_a,l_b))==F){
tot=setdiff(c(a,b),ab)
l_tot=length(tot)
tot=sample(tot, l_tot, replace = FALSE, prob = NULL)
L=l_a-l_ab
hp[[AB[1]]] = c(ab,tot[1:L])
hp[[AB[2]]] = c(ab,tot[(L+1):l_tot])
}
rm=matrix(0,R,C)
for (row in 1:R){rm[row,hp[[row]]]=1}
rm
```

```
}
```

```
####Example
```

```
mat<-matrix(runif(100),10)
```

```
fill<-0.4
```

```
mat1<-1*(mat>fill)
```

```
###species are rows, columns are localities
```

```
###compute species rarity based on percentile
```

```
###number of localities where each species is found
```

```
loc_n<-rowSums(mat1)
```

```
###identify threshold for rarity
```

```
rar_tre<-quantile(loc_n,0.75)
```

```
###non rare species
```

```
nr_spp<-which(loc_n<rar_tre)
```

```
###number rare species per loc
```

```
rar_mat<-mat1
```

```
rar_mat[nr_spp,]<-0
```

```
r_spp_n<-colSums(rar_mat)
```

```
###null model
```

```
nm_sc<-c()
```

```
for (rep in 1:100){
```

```
nm_rar_mat<-FF(mat1)
```

```
nm_rar_mat[nr_spp,]<-0
```

```
nm_sc<-rbind(nm_sc,colSums(nm_rar_mat))
```

```
}
```

```
nm_mean<-apply(nm_sc,2,'mean')
```

```
nm_sd<-apply(nm_sc,2,'sd')
```

```
z<-(r_spp_n-nm_mean)/nm_sd
```

```
#number of significant rare species
```

```
sum(z>2)
```

```
###now explore the effect of matrix fill (~ sampling intensity)
```

```
mat<-matrix(runif(300),30)
```

```

fill_vals<-seq(0.2,0.9,0.01)
hs<-c()
for (fill in fill_vals){
mat1<-1*(mat>fill)
loc_n<-rowSums(mat1)
rar_tre<-quantile(loc_n,0.75)
nr_spp<-which(loc_n<rar_tre)
rar_mat<-mat1
rar_mat[nr_spp,]<-0
r_spp_n<-colSums(rar_mat)
nm_sc<-c()
for (rep in 1:100){
nm_rar_mat<-FF(mat1)
nm_rar_mat[nr_spp,]<-0
nm_sc<-rbind(nm_sc,colSums(nm_rar_mat))
}
nm_mean<-apply(nm_sc,2,'mean')
nm_sd<-apply(nm_sc,2,'sd')
z<-(r_spp_n-nm_mean)/nm_sd
#number of significant rare species
hs<-c(hs,mean(z,na.rm=T)) #explore average z; can be replaced with sum(z>2) for example
}

plot(fill_vals,hs)

```